# A novel stochastic simulation approach enables exploration of mechanisms for regulating polarity site movement

Samuel A. Ramirez [1]*, Michael Pablo [2,3¤], Sean Burk [1], Daniel J. Lew [4], Timothy C. Elston [1]*

**1** Department of Pharmacology and Computational Medicine Program, University of North Carolina at Chapel Hill, Chapel Hill, North Carolina, United States of America, **2** Department of Chemistry, University of North Carolina at Chapel Hill, Chapel Hill, North Carolina, United States of America, **3** Program in Molecular and Cellular Biophysics, University of North Carolina at Chapel Hill, Chapel Hill, North Carolina, United States of America, **4** Department of Pharmacology and Cancer Biology, Duke University, Durham, North Carolina, United States of America

¤ Current address: Gladstone/University of California San Francisco Center for Cell Circuitry, Gladstone Institutes, San Francisco, California, United States of America
* s_a_ramirez@outlook.com (SAR); timothy_elston@med.unc.edu (TCE)

**Data Availability Statement:** All the code to generate data and figures is available from: https://

## Abstract

Cells polarize their movement or growth toward external directional cues in many different contexts. For example, budding yeast cells grow toward potential mating partners in response to pheromone gradients. Directed growth is controlled by polarity factors that assemble into clusters at the cell membrane. The clusters assemble, disassemble, and move between different regions of the membrane before eventually forming a stable polarity site directed toward the pheromone source. Pathways that regulate clustering have been identified but the molecular mechanisms that regulate cluster mobility are not well understood. To gain insight into the contribution of chemical noise to cluster behavior we simulated clustering using the reaction-diffusion master equation (RDME) framework to account for molecular-level fluctuations. RDME simulations are a computationally efficient approximation, but their results can diverge from the underlying microscopic dynamics. We implemented novel concentration-dependent rate constants that improved the accuracy of RDME-based simulations, allowing us to efficiently investigate how cluster dynamics might be regulated. Molecular noise was effective in relocating clusters when the clusters contained low numbers of limiting polarity factors, and when Cdc42, the central polarity regulator, exhibited short dwell times at the polarity site. Cluster stabilization occurred when abundances or binding rates were altered to either lengthen dwell times or increase the number of polarity molecules in the cluster. We validated key results using full 3D particle-based simulations. Understanding the mechanisms cells use to regulate the dynamics of polarity clusters should provide insights into how cells dynamically track external directional cues.

github.com/samuramirez/stochastic-exploratory-polarization.

**Funding:** This work was supported by NIH/NIGMS grants R35GM122488 to D.J.L and R35GM127145 to T.C.E. The funders had no role in study design, data collection and analysis, decision to publish, or preparation of the manuscript.

**Competing interests:** The authors have declared that no competing interests exist.

## Author summary

Cells localize polarity molecules in a small region of the plasma membrane forming a polarity cluster that directs functions such as migration, reproduction, and growth. Guided by external signals, these clusters move across the membrane allowing cells to reorient growth or motion. The polarity molecules continuously and randomly shuttle between the cluster and the cell cytosol and, as a result, the number and distribution of molecules at the cluster constantly changes. Here we present an improved stochastic simulation algorithm to investigate how such molecular-scale fluctuations induce cluster movement across the cell membrane. Unexpectedly, cluster mobility does not correlate with variations in total molecule abundance within the cluster, but rather with changes in the spatial distribution of molecules that form the cluster. Cluster motion is faster when polarity molecules are scarce and when they shuttle rapidly between the cluster and the cytosol. Our results suggest that cells control cluster mobility by regulating the abundance of polarity molecules and biochemical reactions that affect the time molecules spend at the cluster. We provide insights into how cells harness random molecular behavior to perform functions important for survival, such as detecting the direction of external signals.

## Introduction

Cell migration, division, and differentiation require breaking the internal symmetry of the cell and establishing an axis of orientation. This symmetry breaking is referred to as polarity establishment. In eukaryotes, polarity establishment occurs as polarity factors, such as Rho-family GTPases, localize in a small region of the plasma membrane where they regulate the cytoskeleton to remodel cell morphology and generate motility [1]. In particular contexts, the polarity site can be highly dynamic. For example, migrating cells frequently change their direction of polarization as they navigate guided by changing environmental cues [2–4].

   Polarity establishment has been well characterized in the budding yeast *Saccharomyces cerevisiae*. Yeast polarize in the contexts of budding and mating. The first step involves the clustering of the conserved master regulator of polarity, the Rho-GTPase Cdc42, at a site on the plasma membrane often referred to as the "polarity patch". In the context of mating, detection of pheromone secreted by a potential mating partner can trigger polarization. However, the location of the initial polarity patch is inaccurate, and often misaligned with respect to the pheromone source [5,6]. The patch then relocates so that it is adjacent to a neighboring mating partner, allowing the two cells to fuse. Relocation of the polarity patch occurs in two stages. Initially the polarity patch is highly dynamic, rapidly assembling, disassembling, and moving along the cell membrane. In the next stage, Cdc42 organizes into a more concentrated patch with reduced mobility [5,6]. The initial rapid movement of the polarity patch is thought to be an exploratory phase to locate a mating partner. The remaining mobility of the patch during the second stage may be necessary to correct errors made during the exploratory phase. This view is supported by the observation that in experiments using externally imposed pheromone gradients, cells that did not polarize toward the gradient during the exploratory phase were able to reorient the polarity patch in the direction of the gradient [7–12]. Investigations combining experimental studies with mathematical modeling showed that actin-based vesicle delivery to the polarity patch is a key driver of patch movement during the second stage [7,13,14]. However, the mechanisms responsible for generating highly dynamic clustering during the exploratory phase and the transition to more stable polarity at the end of this stage are not well understood.

Recent studies revealed that the mobility of the polarity patch during mating is correlated with MAPK activity [5,6]. Pheromone-induced MAPK activity triggers polarization and drives changes in gene expression required for mating. During the early phases of mating when the polarity patch is highly mobile, MAPK activity is low. As the dynamic cluster of Cdc42 explores the membrane and relocates to a region near a mating partner, MAPK activity increases and the cluster of Cdc42 at the membrane becomes stable. Hegemann et al. [5] proposed that MAPK activity regulates patch mobility by inducing nuclear export of Cdc24, the GEF (activator) for Cdc42, thereby increasing Cdc42 activation at the membrane. They also proposed that stochastic fluctuations in the biochemical events underlying Cdc42 polarization drive the mobility of the cluster during the exploratory phase. The plausibility of the second claim was supported using a simple stochastic model for cell polarization adapted from [15]. Their mathematical formulation, however, did not address the mechanism for cluster stabilization.

To gain further insight into how chemical noise can induce cluster mobility and how cells can regulate cluster dynamics, we considered mechanistically detailed stochastic models of cell polarization. In a previous study, we used particle-based simulations to demonstrate that molecular-level fluctuations favor polarity establishment [16]. The stochastic simulations resulted in an extended parameter range over which polarity occurs and shorter times (1–5 min) for the emergence of a single polarity site in comparison to a deterministic reaction-diffusion version of the model. However, because particle-based simulations are computationally expensive, we were not able to address the stochastic behavior of the polarity patch over the time scales (10–100 min) associated with patch movement during yeast mating. Therefore, there is a need for efficient approximate methods that faithfully capture intrinsic fluctuations and allow simulations to be performed over biologically relevant time scales.

The reaction-diffusion master equation (RDME) provides an approximate method for describing stochastic reaction-diffusion dynamics. In this framework, space is discretized into a grid of volume elements which are assumed to be "well-stirred". Within grid elements reactions occur with a propensity proportional to rate constants usually referred to as "mesoscopic rate constants". This terminology is used to distinguish them from rate constants that appear in macroscopic chemical rate equations derived under the assumption of mass action kinetics. While solutions of the full RDME are typically difficult to achieve, computer simulations performed with an optimized version of the Gillespie algorithm can be used to efficiently generate realizations of the system's spatiotemporal dynamics [17]. A limitation of the RDME approach is that it is challenging to find mesoscopic parameters that faithfully capture the underlying microscopic dynamics. Several modifications to the RDME approach have been developed to overcome this shortcoming. These improved methods rely on mesoscopic rate constants that take into account the finite size of the grid elements used to discretize space [18–20]. However, we demonstrate that even with these improvements, RDME-based simulations lose accuracy at the high molecular densities typical of the polarity system. To overcome this limitation, we derived concentration-dependent mesoscopic rate constants, extending the work of Yogurtcu et al [21]. We validated our approach in a 2D geometry by comparisons with particle-based simulations. We then applied our approach to study the effects of molecular-level fluctuations on the mobility of the polarity site in yeast. Key results were confirmed using full 3D particle-based simulations.

We found that molecular-level fluctuations can induce high mobility in Cdc42 clusters when clusters contain low numbers of the GEF for Cdc42 (the limiting polarity factor) and Cdc42 rapidly cycles between the cluster and the cytosol. Cluster stabilization was observed when GEF abundance increased or when the rate constant for association reactions between membrane-bound molecules increased. Accelerating such reactions stabilized clusters mainly by increasing the dwell time of Cdc42 or GEF molecules at the polarity patch. Interestingly,

increasing the rate constant for formation of the Cdc42-GEF complex, which is involved in the positive feedback for Cdc42 activation, produced a switch-like transition in patch dynamics, suggesting its regulation may underlie patch stabilization during yeast mating.

## Results

### A. An improved approach for stochastic reaction-diffusion simulations

**A.1. Background.**   At the microscopic level, second-order reactions can be thought of as occurring in two steps. First, the two reactants must diffuse to close enough proximity for the reaction to occur. Second, after the reactants encounter one another, there may still be enthalpic and entropic barriers to overcome before the reaction can proceed. Second-order reactions are diffusion-limited when, the "search" time for two reactants to encounter one another is substantially longer than the time for the reaction to proceed once the reactants are close enough to interact. Conversely, the reactions are reaction-limited when the diffusional search time is short in comparison to the reaction time.

Particle-based simulations that track the position of each molecule in the system provide a microscopic representation of chemically reacting systems (Fig 1A). Typically, in these simulations it is assumed that once the reactants meet, the reaction occurs instantaneously [22,23] or requires a single kinetic step to proceed [24–26] (see Methods for a discussion of microscopic models for second-order reactions). For the particle-based simulations presented here, we followed an implementation of the latter approach in which molecules react with a rate $\lambda$ when their separation is within an interaction radius $\rho$ [26]. Therefore, the microscopic parameters required for simulating second-order reactions are the diffusion coefficients for the reacting molecular species, $\rho$ and $\lambda$. Simulating first order reactions, such as the dissociation of molecular complexes, requires specifying a single rate constant for the reaction. For particular cases, mathematical expressions that relate the microscopic parameters to experimentally measured macroscopic rate constants (i.e., rate constants that appear in mass action kinetics) have been derived [21,24,26]. Such mathematical expressions can be used to parameterize particle-based simulations that are consistent with experimental measurements.

Microscopically-detailed, particle-based simulations are typically computationally expensive. Therefore, to reduce simulation times it is common to use an approximate description of the system based on the reaction-diffusion master equation (RDME). In this approach, space is discretized into a grid of volume elements and molecules are assumed to be "well-mixed" within the volume elements (Fig 1B). This is a mesoscopic description in the sense that the size of the grid elements is normally larger than the size of a molecule (microscopic scale), but still significantly smaller than the whole system (macroscopic scale). Molecules can jump to adjacent grid elements with a propensity proportional to the diffusion coefficient and reactions occur with a propensity proportional to the reactants' abundances within the grid element and a mesoscopic rate constant $k_{\mathrm{meso}}$. As the exact solution of the RDME can be difficult to obtain, realizations of the system are typically simulated with an efficient spatial version of the Gillespie algorithm [17].

A challenge with RDME simulations is determining parameters that produce results consistent with the underlying microscopic dynamics. This can be especially problematic when second-order reactions are diffusion-limited because simulation results can strongly depend on the grid spacing $h$ [27]. This issue was addressed recently through the derivation of mesoscopic rate constants in terms of microscopic parameters that ensure compliance with the microscopic kinetics in the limit of low molecular abundances [18–20]. Specifically, this approach considered a domain containing two diffusing molecules that undergo an association reaction. A mesoscopic rate constant is then derived using the requirement that the mean association

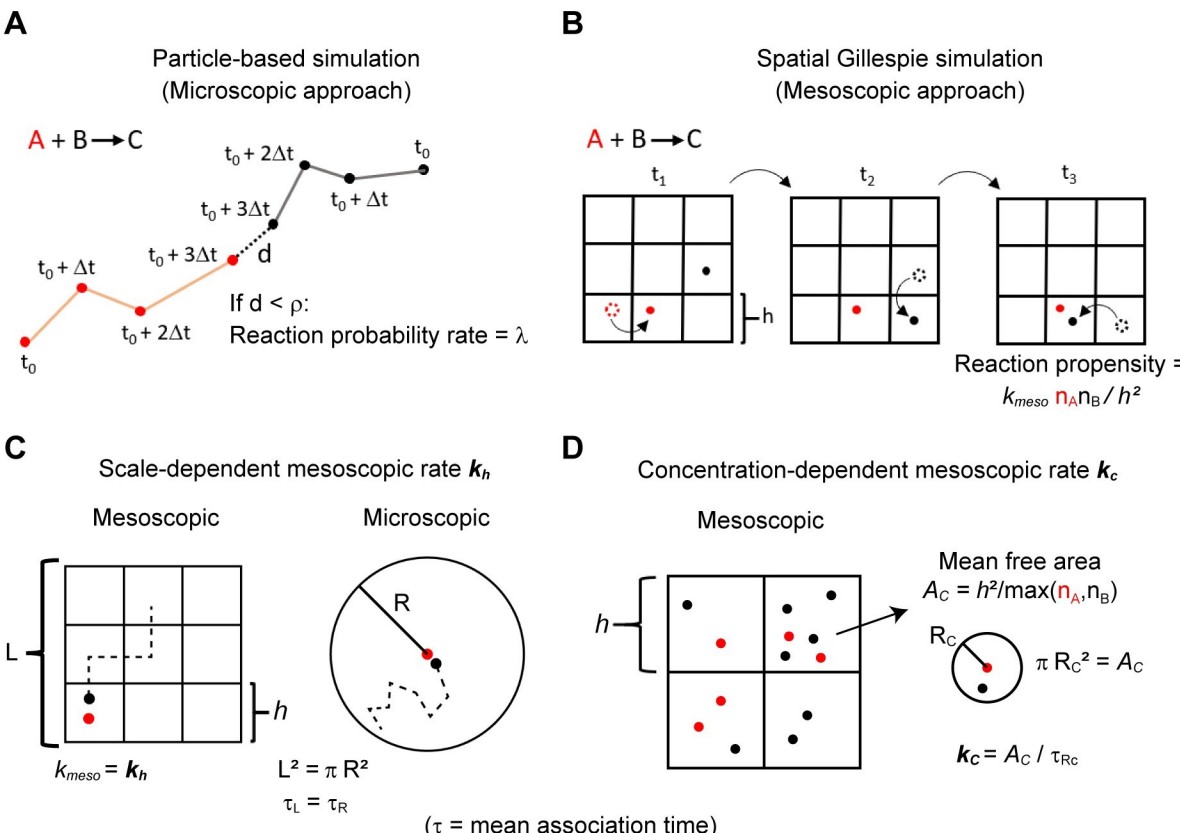

**Fig 1. Simulation of a bimolecular reaction using a particle-based method and the spatial Gillespie approach.** In our particle-based simulations (**A**) molecules undergo a random walk in continuous space and discrete time intervals $\Delta t$. If a pair of reacting molecules are within a distance $\rho$, they can react with probability rate $\lambda$. In the spatial Gillespie approach (**B**), the domain is discretized using a grid, here with square elements of size $h$. Molecule jumps to adjacent grid elements and reactions take place within grid elements at random times. The propensity of a reaction within a grid element is proportional to the number of molecules ($n_A$, $n_B$) and a mesoscopic rate constant $k_{meso}$. (**C**) The scale-dependent mesoscopic rate $k_h$ ensures that the mean association time of two molecules in the reaction-diffusion master equation ($\tau_c$) matches an analogous microscopic representation ($\tau_R$). (**D**) A concentration-dependent mesoscopic rate constant is defined as $k_c = A_c/\tau_{Rc}$ where $A_c$ is the mean free area between molecules of the most abundant reactant in the grid element, estimated as $A_c = h^2/\max(n_A, n_B)$, and $\tau_{Rc}$ is the mean association time calculated from a microscopic model of two molecules that react in a circular domain with area $\pi R_c^2 = A_c$. See Methods section for further details.

time for the mesoscopic description is equivalent to that of the microscopic representation (Fig 1C, Methods). Because this mesoscopic rate constant depends on $h$, it has been referred to as a "scale-dependent" rate constant. If the association reaction is reversible, a mesoscopic dissociation rate constant is computed by ensuring that the equilibrium behavior of the mesoscopic description is identical to that of the microscopic representation (Methods).

In Section A.2. we evaluate the scale-dependent mesoscopic rate $k_h$ derived in [19,20] by comparing simulations of simple reaction schemes using $k_h$ with results from particle-based simulations. We note that although $k_h$ yields accurate results at low concentrations, it shows significant deviations from particle-based simulations in scenarios with high molecular abundances. To address this issue, in Section A.3. we propose a concentration-dependent mesoscopic rate $k_c$ that produces accurate results over a broad range of concentrations. In Section A.4 we evaluate both $k_h$ and $k_c$ in a biochemical model for polarity establishment in yeast.

**A.2. Evaluation of the scale-dependent mesoscopic rate $k_h$.** We evaluated the scale-dependent mesoscopic rate $k_h$ [19,20] by considering two prototypical reactions: irreversible

association A+B → C and reversible association A+B ↔ C. A measure of the degree to which the reaction is controlled by diffusion is the dimensionless ratio $\lambda\pi\rho^2/(D_A+D_B)$ where $D_A$ and $D_B$ are the diffusion coefficients of the A and B molecules respectively. For the results presented in this section, we set the value of this ratio to 50 to ensure the reactions are strongly controlled by diffusion. Because our goal is to model reaction and diffusion at the cell membrane, simulations are performed in a 2D domain with periodic boundary conditions.

To test the accuracy of $k_h$, we performed simulations in which either the size of computational domain was fixed and the molecular abundances varied (Fig 2A–2D) or the molecular abundances fixed and the size of the computational domain varied (S1 Fig). For each case, we present time series for the mean number of molecules of one of the reactants. At low reactant concentrations, simulations using $k_h$ are consistent with the microscopic dynamics as expected (Figs 2A and 2B and S1G–S1J) [19,20]. We observed small differences for the reversible reaction that may be attributed to differences in the assumptions used to derive $k_h$ and the method we used to perform particle-based simulations. At higher concentrations, however, the early kinetics of both the reversible and irreversible reactions showed significant deviations from the particle-based results (Figs 2C and 2D and S1A–S1F). At later times the deviations are reduced as the number of molecules decreases. We note that it is not possible to achieve higher accuracies by reducing the size of the grid elements, because for this parameter regime, it is not possible to calculate $k_h$ for $h$ smaller than ~5ρ (Methods) [19,20].

We also computed timeseries of the standard deviation to quantify the fluctuations around the mean for each case in Fig 2A–2D (S2A–S2D Fig). This metric also showed agreement between spatial Gillespie simulations using $k_h$ and particle-based simulations at low concentrations, and increased deviations at high concentrations at early times. Overall, the mesoscopic approach using $k_h$ provides a good approximation to the microscale dynamics for irreversible and reversible diffusion-controlled reactions only if the density of reactants is low.

To understand why $k_h$ showed accurate results for systems with low abundances but loses accuracy at high concentrations, remember that $k_h$ was derived to reproduce the association time in a two-molecule system (Fig 1C, Methods). In a dilute system containing $n$ pairs of reacting molecules, the expected time for an association event is similar to that in a set of $n$ independent 2-molecule systems, and simulations using $k_h$ provide a good approximation of the kinetics. At high concentrations, however, molecules are more likely to associate with partners in their vicinity before diffusing over a significant portion of the spatial domain. Because the derivation of $k_h$ does not consider such multi-molecule effects, simulations lose accuracy at higher concentrations.

**A.3. A concentration-dependent mesoscopic rate $k_c$ improves accuracy at high concentrations.** As the concentration of reactants increases, the reaction rate becomes dominated by the time for reactants within the same grid element to react, as opposed to the time for two reactants to diffuse into the same grid element. Therefore, to improve accuracy at increased concentrations with diffusion-controlled reactions, we applied a mesoscopic rate constant $k_c$ that describes the kinetics within a grid element without assuming that the volume element is "well-stirred". This mesoscopic rate constant considers that a molecule's mean-free-path before reaction decreases with the concentration of its reacting partner. $k_c$ is defined as $A_c/\tau_{Rc}$, where $\tau_{Rc}$ is the mean association time for two molecules diffusing in a domain with area $A_c$, with this area taken as the mean free area between molecules of the more abundant reactant within the grid element (Fig 1D). As an estimate for $A_c$ we use the grid element area $h^2$ divided by the number of molecules of the more abundant reactant. We present a derivation for $k_c$ in the Methods section. This approach was inspired by the work of Yogurtcu et al [21], in which the authors estimated a concentration-dependent rate constant for spatially homogeneous

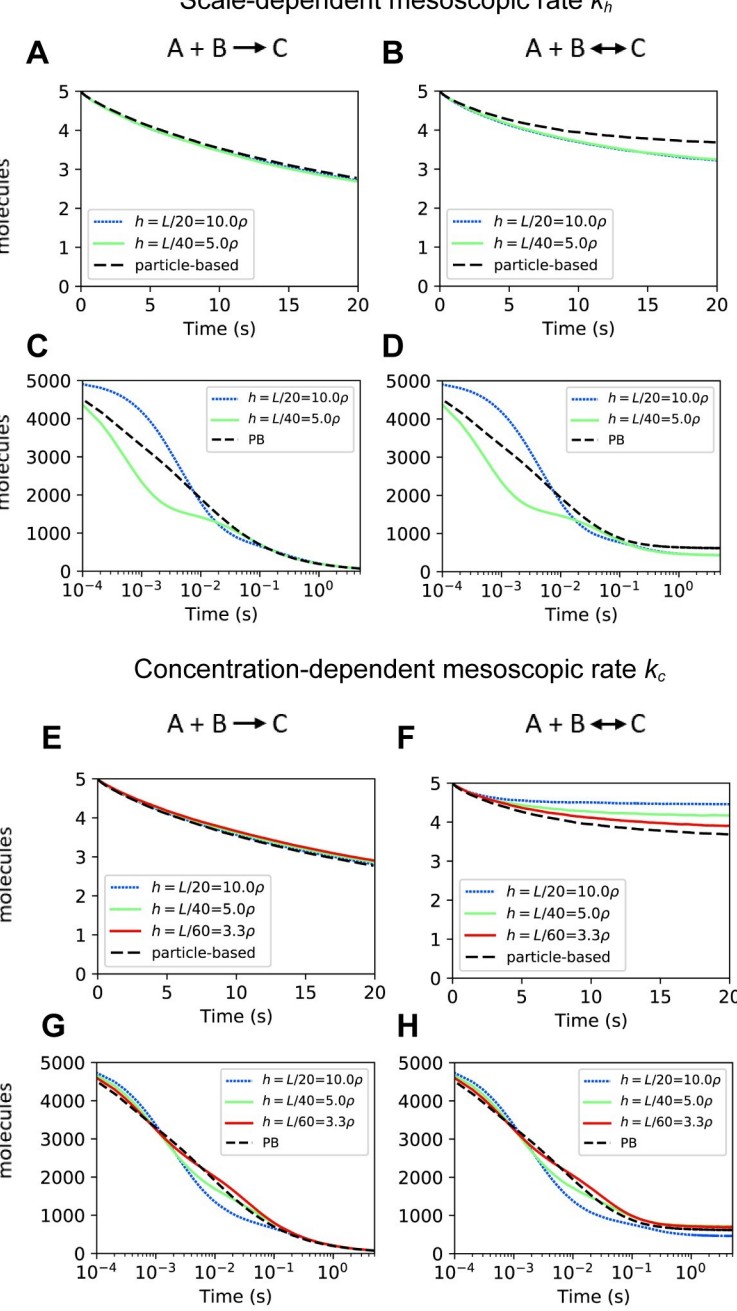

**Fig 2. Comparison of spatial Gillespie simulations using the mesoscopic rates $k_h$ and $k_c$ with particle-based simulations. (A-D)** Results for the mean number of species A from spatial Gillespie simulations using the mesoscopic rate $k_h$ with initial low abundance of reactants in **(A, B)** (total A = total B = 5, total C = 0 at t = 0) and initial high abundance (total A = total B = 5000, total C = 0 at t = 0) in **(C-D)**. **(E-H)** show corresponding simulations to **(A-D)** but using the mesoscopic rate $k_c$. In all the simulations, the degree of diffusion control is $\lambda\pi\rho^2/D_{tot} = 50$, with $D_{tot} = 2D$ and $D = 0.0025\mu m^2/s$, $\rho = 0.005\ \mu m$ and $\lambda = 3183.1/s$. The size of the domain is $L = 1\mu m$. For the reversible reaction A+B $\leftrightarrow$ C, the microscopic dissociation rate constant $k^d_{micro}$ is 1/s in **(B, F)**, and 10/s in **(D, H)**.

systems. In our simulation approach, $k_c$ depends on the local concentration and therefore it changes in space and time as the system evolves.

For the case of reversible second-order reactions, a concentration-dependent mesoscopic dissociation rate $k_c^d$ is estimated in a similar way as for $k_h^d$. That is, the equilibrium behavior of a two-species system in the mesoscopic representation is matched to that of the microscopic formulation for a pair of molecules diffusing in a domain with area $A_c$ (Methods).

Even though $k_c$ was derived to provide accurate results at high concentrations, it maintained accuracy for low molecular abundances in the reaction A + B → C (Figs 2E and S1G–S1J). In the reversible reaction A+B ↔ C, simulations showed deviations that decreased with smaller grid element size $h$ (Fig 2F). At high molecular abundance, $k_c$ showed increased accuracy compared to $k_h$ in both irreversible and reversible reactions (compare Fig 2C–2D with Fig 2G–2H, also see S1A–S1F Fig). The increase in accuracy using $k_c$ was also observed in the fluctuations around average concentrations (compare S2C–S2D Fig with S2G–S2H Fig).

**A.4. Evaluation of mesoscale simulations in a model of the yeast polarity circuit.** It is known that the RDME approach can generate anomalous behavior due to numerical artifacts from using a finite grid [28]. Therefore, we compared results from the mesoscale rates $k_h$ and $k_c$, and particle-based simulations, in a reaction-diffusion model for polarity establishment in budding yeast adapted from [29] (Fig 3A). Central to this biochemical network is the Rho-GTPase Cdc42. Cdc42 can exist in an inactive (GDP bound) form Cdc42D that shuttles between the membrane and the cytosol, and an active (GTP bound) form Cdc42T that localizes to the cell membrane. Deactivation occurs as Cdc42 hydrolyzes GTP into GDP, a process accelerated by GTPase activating proteins (GAPs). GAP activity is considered implicitly using a pseudo-first order deactivation rate constant. The activation of Cdc42 is catalyzed by the guanine nucleotide exchange factor Cdc24 (GEF) that binds Cdc42D and facilitates the exchange of GDP for GTP. GEF molecules can shuttle between membrane and cytosol, and at the membrane they activate Cdc42. Once GEF binds Cdc42D, it activates the Rho-GTPase and dissociates from the resulting Cdc42T in a single step. There is positive feedback in the levels of active Cdc42 because Cdc42T can bind a GEF molecule forming a complex Cdc42T-GEF that can activate neighboring Cdc42D molecules. In this model Cdc42T can recruit both membrane-bound and cytosolic GEF molecules. In the following simulations, membrane and cytosol are represented as coincident 2D square domains. What distinguishes the membrane and cytosol are the diffusion coefficients for the molecular species in each domain. The parameters were adapted from a 3D macroscopic model [13,29] into a 2D microscopic representation (Methods) and are presented in Table 1.

Particle-based simulations initialized with all Cdc42 and GEF molecules randomly located in the cytosol evolved into a polarized distribution of total Cdc42T (Cdc42T and Cdc42T-GEF) at the membrane (Fig 3B) [16]. At early times, small fluctuations in the levels of Cdc42T are amplified by positive feedback reactions resulting in growing clusters of the active Rho-GTPase (Fig 3B 10 s). Several clusters can form that compete for polarity factors in the cytosol until just one cluster remains (Fig 3B, 60 s—120 s). The remaining cluster grows, depleting polarity factors in the cytosol, until a steady state is reached when the inward and outward fluxes of molecules to the cluster balance (Fig 3B 120 s– 300 s).

In Fig 3C and 3D we present snapshots of representative mesoscopic simulations using the scale-dependent ($k_h$) and concentration-dependent ($k_c$) rates, respectively. To facilitate comparison with particle-based simulations, spatial pseudo-coordinates for each molecule were obtained by randomly sampling within the grid element containing the molecule. Mesoscopic simulations using both $k_h$ and $k_c$ reached a steady-state polarity cluster (Fig 3C and 3D 300 s) with a size comparable to that of the particle-based simulation in Fig 3A. There were differences, however, in the time evolution of polarization between the different methods. The simulation using $k_h$ took longer to evolve into a single polarity cluster, while the simulation using $k_c$

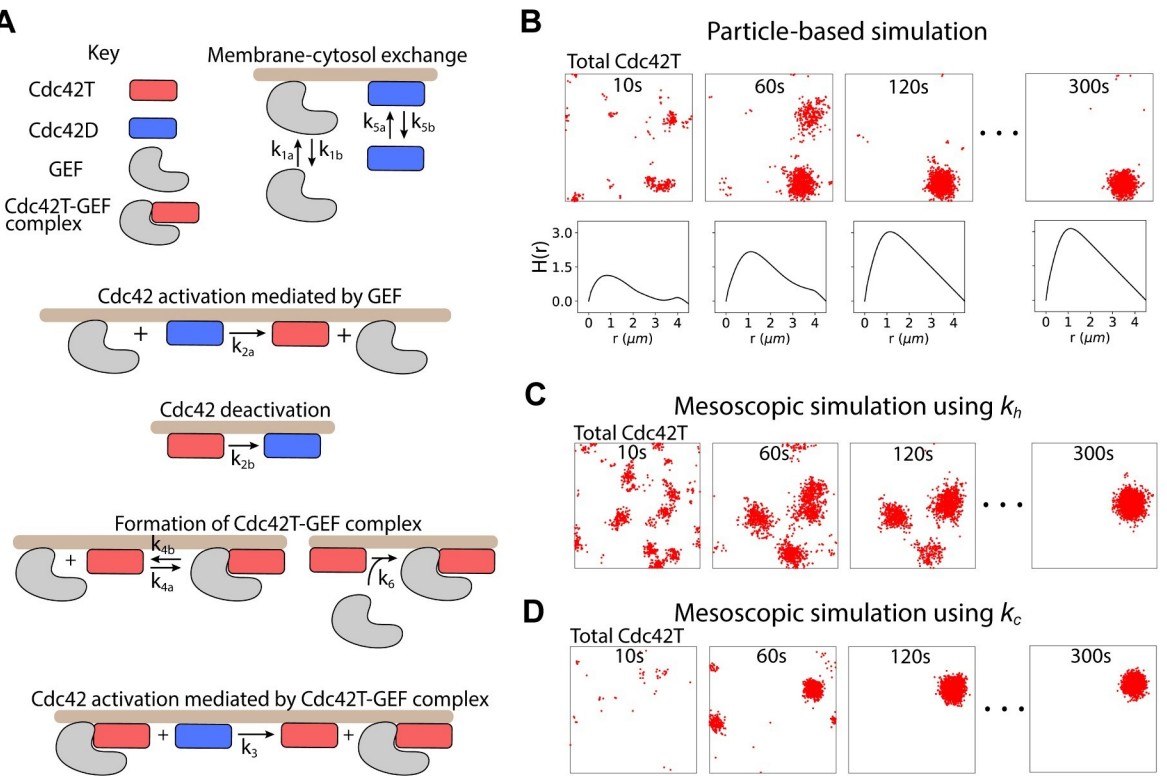

**Fig 3. Cdc42 polarization simulated with particle-based simulations and spatial Gillespie simulations using $k_h$ and $k_c$. (A)** Reactions in a model for polarization in budding yeast. Species that are not bound to the membrane (brown), dwell in the cytosol. Membrane and cytosol are represented in the simulations as juxtaposed 2D squared domains. **(B)** Time series of a particle-based simulation of the polarization model in **(A)**. In each snapshot, red dots show the positions on the membrane of all active Cdc42 molecules (Cdc42T and Cdc42T-GEF). The lower panels in **(B)** show a quantification of active Cdc42 clustering using the H($r$) function (see Methods for details). **(C, D)** Spatial Gillespie simulations using $k_h$ **(C)** and $k_c$ **(D)** with the same model parameters as the particle-based simulation in **(A)** and grid element size h = 5ρ. Pseudo-coordinates for each molecule are randomly sampled from the containing grid element and displayed as a red dot to facilitate comparison with particle-based simulations. Model parameters are presented in Table 1.

polarized faster compared to the particle-based approach. We note that these simulations were run using a grid element $h = 5ρ$ which corresponds to the finest grid possible using $k_h$.

To quantify the dynamics of polarization we measured clustering of total Cdc42T with the function H($r$) (Fig 3B lower panels) [16,30,31] (Methods). H($r$) has the desired properties that H($r$) = 0 for a random distribution of molecules, and for a clustered distribution, H($r$) shows a maximum at a value $r = r_{max}$ which provides a measure of the cluster size. H($r$) showed a maximum at $r = 1.1$ μm that increased over time (Fig 3B lower panels). We therefore chose H($r = 1.1$ μm) as a metric for polarization. We note that quantifying clustering with max H($r$) does not qualitatively change the results. In Fig 4A we show the time evolution of the mean of H($r = 1.1$ μm) over different realizations for the different simulation methods. The shading indicates standard deviation to illustrate the variability in polarization dynamics. At early times, simulations using $k_h$ matched particle-based simulations but showed deviations at later times, taking longer to polarize. On the other hand, simulations using $k_c$ displayed overall faster polarization than the other methods. By defining the polarization time as the moment when the standard deviation stabilizes, it is apparent that simulations using $k_h$ have a longer time of polarization (240 s) compared to particle-based simulations (180 s), while using $k_c$ results in faster polarization (130 s).

**Table 1. Model parameters.** For reactions, the values listed are for microscopic rate constants. For second-order reactions, the corresponding λ values can be calculated from Eqs 2 or 26, depending on the type of reaction. The parameters $k_h$, $k_h^d$, $k_c$ and $k_c^d$ used in spatial Gillespie simulations are computed from Eqs 8, 13, 16 and 19, respectively. The parameters for the 3D model correspond with the 2D parameters, except when they have been scaled to account for dimensionality (Methods). Both 2D and 3D particle-based simulations were performed using Smoldyn [22,23] specifying reaction probabilities for second-order interactions. Such probabilities are computed from the rate constants presented here as described in the Methods.

| Description | Param. | 2D Model | 3D Model |
|---|---|---|---|
| $GEF_c \rightarrow GEF_m$ | $k_{1a}$ | 0.1 s$^{-1}$ | 0.07522 µm s$^{-1}$ |
| $GEF_m \rightarrow GEF_c$ | $k_{1b}$ | 10 s$^{-1}$ | 10 s$^{-1}$ |
| $Cdc42D_m + GEF_m \rightarrow GEFm + Cdc42T$ | $k_{2a}$ | 0.032 µm$^2$s$^{-1}$ | 0.032 µm$^2$s$^{-1}$ |
| $Cdc42T \rightarrow Cdc42D_m$ | $k_{2b}$ | 0.63 s$^{-1}$ | 0.63 s$^{-1}$ |
| $Cdc42D_m + Cdc42T\text{-}GEF \rightarrow Cdc42T\text{-}GEF + Cdc42T$ | $k_3$ | 0.07 µm$^2$s$^{-1}$ | 0.07 µm$^2$s$^{-1}$ |
| $GEF_m + Cdc42T \rightarrow Cdc42T\text{-}GEF$ | $k_{4a}$ | 0–2 µm$^2$s$^{-1}$ | 0–2 µm$^2$s$^{-1}$ |
| $Cdc42T\text{-}GEF \rightarrow GEF_m + Cdc42T$ | $k_{4b}$ | 10 s$^{-1}$ | 10 s$^{-1}$ |
| $Cdc42D_c \rightarrow Cdc42D_m$ | $k_{5a}$ | 4 s$^{-1}$ | 3.009 µm s$^{-1}$ |
| $Cdc42D_m \rightarrow Cdc42D_c$ | $k_{5b}$ | 6.5 s$^{-1}$ | 6.5 s$^{-1}$ |
| $GEF_c + Cdc42T \rightarrow Cdc42T\text{-}GEF$ | $k_6$ | 0.2 µm$^2$s$^{-1}$ | 0.15 µm$^3$s$^{-1}$ |
| $Cdc42D_c + Cdc42T\text{-}GEF \rightarrow Cdc42T\text{-}GEF + Cdc42T$ | $k_7$ | 0.5 µm$^2$s$^{-1}$ | 0.376 µm$^3$s$^{-1}$ |
| Diffusion coefficient in cytoplasm | $D_{cyto}$ | 10 µm$^2$s$^{-1}$ | 10 µm$^2$s$^{-1}$ |
| Diffusion coefficient on membrane | $D_{memb}$ | 0.0045 µm$^2$s$^{-1}$ | 0.0045 µm$^2$s$^{-1}$ |
| Membrane surface area | $A_m$ | 64 µm$^2$ | 64 µm$^2$ |
| Total Cdc42 | Cdc42 | 5000 molecules | 5000 molecules |
| Total GEF | GEF | 15–700 molecules | 15–700 molecules |
| Membrane thickness | Δz | 0.0083 µm | 0.0083 µm |
| Cell volume | $V_c$ | 48.144 µm$^3$ | 48.144 µm$^3$ |
| Membrane volume | $V_m$ | 0.53 µm$^3$ | 0.53 µm$^3$ |
| $V_m/V_c$ | η | 0.011 | 0.011 |
| Reactive radius | ρ | 0.02 µm | 0.02 µm |

We further characterized the simulation approaches by looking at the steady state behavior for different amounts of available GEF molecules (Fig 4B). While particle-based simulations lost polarity at GEF amounts of around 300 molecules, mesoscopic simulations using $k_h$ polarized for GEF abundances of around 100, showing polarization in a regime where the microscopic simulations do not spontaneously polarize. On the other hand, mesoscopic simulations using $k_c$ showed polarization for GEF amounts greater than 400, failing to polarize at a value of 400 GEF where the microscopic approach shows polarization.

As the mesoscopic simulations with both $k_h$ and $k_c$ presented in Fig 4A and 4B showed discrepancies with respect to particle-based simulations, we sought to obtain more accurate results by reducing the grid element size, which so far was set to h = 5ρ. We were able to do this only for simulations with the concentration-dependent rate $k_c$, because for the parameters of the model, $k_h$ cannot be computed for grid elements smaller than 5ρ (see Methods). For h = 2.5ρ we observed a significant improvement in accuracy using $k_c$ both in the dynamics of polarization (Fig 4C) and in the equilibrium behavior for different values of the number of GEF molecules (Fig 4D).

The increase in accuracy using h = 2.5ρ came with a cost of ≈ 5X increase in computation time with respect to simulations using h = 5ρ. Particle based simulations, on the other hand, were ≈ 10X more computationally expensive with respect to mesoscopic simulations using h = 5ρ. We note that particle-based simulations were performed with the highly optimized simulation platform Smoldyn [22,23], whereas mesoscopic simulations were run using our own custom written C code, which has not been optimized for computational performance.

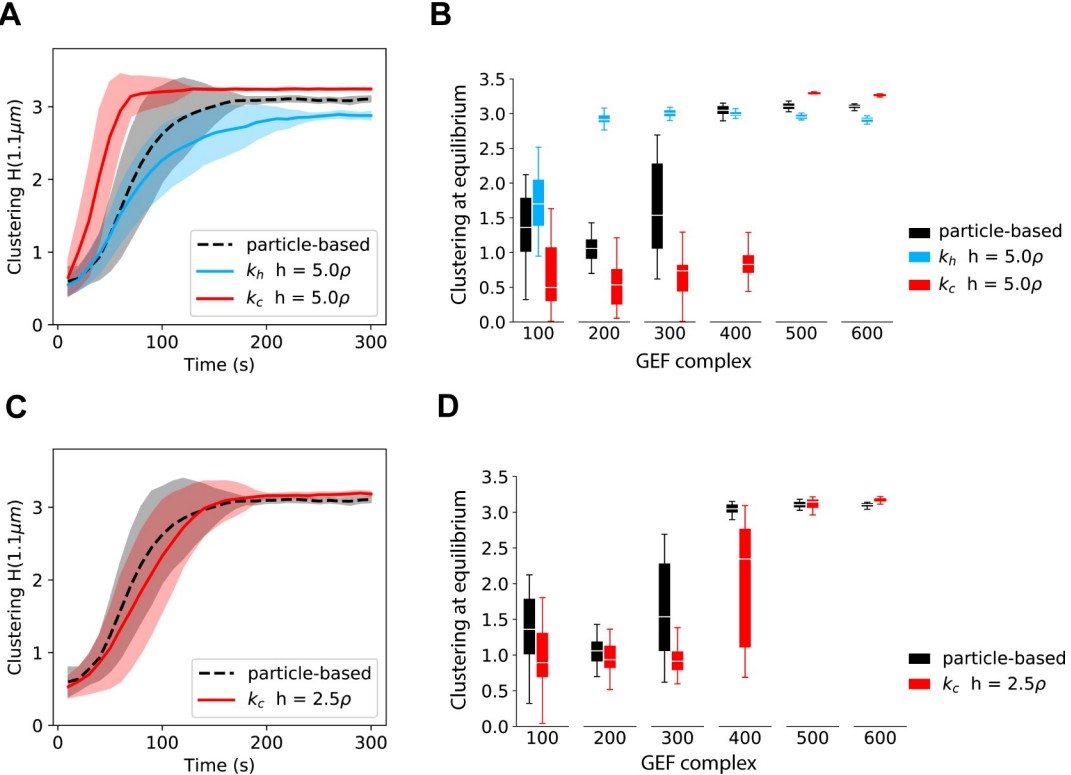

**Fig 4. $k_h$ and $k_c$ are benchmarked against particle-based simulations by looking at polarization dynamics and equilibrium.** **(A)** Time evolution of the clustering of total active Cdc42 from simulations of the polarity model in Fig 3A with parameters in Table 1. We contrast results from particle-based simulations and the spatial Gillespie approach using either $k_h$ or $k_c$ and a grid element size $h = 5\rho$. Clustering at a particular timepoint is quantified as the mean of H($r = 1.1\mu m$) over 30 simulations. Uncertainty intervals are computed as mean ± standard deviation and presented as a shaded region enveloping the mean. **(B)** Clustering at equilibrium from simulations in **(A)** for different values of the total amount of GEF. The clustering at equilibrium is computed as the mean of H($r = 1.1\mu m$) between 250s and 300s. The box plots were generated with the clustering at equilibrium from 30 simulations. **(C)** and **(D)** are corresponding figures to **(A)** and **(B)** respectively, the only difference is that the spatial Gillespie simulations are run with $h = 2.5\rho$ and using only $k_c$ as $k_h$ cannot be computed for such $h$ in this model (see Methods).

In summary, we found that the mesoscopic rate $k_c$ provided more accurate results than $k_h$ in simulations of elementary reactions at high concentrations. During simulations of the polarity model, both high and low concentrations of reactants can exist simultaneously. Therefore, it is harder to definitively establish which mesoscopic rate provides a better description of the system. We favor the concentration-dependent rate $k_c$ for several reasons: 1) it shows more accurate results at the higher concentrations found in the polarity site, 2) it does not artificially generate polarization in parameter regimes where particle-based simulations do not polarize (although it fails to show polarization for some parameters where the particle-based simulations do polarize), and 3) $k_c$ allows the use of smaller grid elements producing increased accuracy.

## B. Mechanisms regulating mobility of the polarity site

**B.1. Requirements for high patch mobility.** When yeast cells are presented with pheromone from a potential mating partner, Cdc42 forms dynamic clusters that explore the membrane for 10–100 min before stabilizing in a region close to the pheromone source [5,6]. Therefore, we sought to determine if intrinsic fluctuations are sufficient to explain patch

mobility by performing spatial Gillespie simulations using our new concentration-dependent mesoscopic rate constants for bimolecular reactions.

To assess patch mobility, we tracked the centroid of the distribution for active Cdc42 over time (Fig 5A). However, with our initial parameterization (Table 1), the patch did not move significantly over a 60 min time interval. It has been suggested that the amount of GEF in the cytosol strongly regulates cluster dynamics [5]. In previous modeling studies, GEF abundances ranged from 500–2000 molecules [13,29,32,33]. These studies primarily focused on situations where the patch was stable. However, immediately following exposure to pheromone, when the polarity clusters are highly mobile, it is possible that the amount of available GEF is significantly smaller. Therefore, we ran simulations with GEF levels of 700 molecules and fewer. To quantify patch movement, we computed the mean squared displacement (MSD) of the patch centroid over time and used these values to estimate an effective diffusion coefficient $D_{patch}$ (Fig 5B). As anticipated, decreasing GEF abundance increased patch mobility, but the system lost polarity at moderate GEF levels ($\approx 450$ molecules) before patch movement increased substantially. Therefore, we investigated if varying any of the rate constants would allow the system to polarize at lower GEF abundances (S3 Fig). After testing all the reactions in the model (S3 Fig), we observed that the following modifications allowed the system to polarize with low GEF abundances (100 or fewer molecules): 1) decreasing the rate constant $k_{2b}$ for Cdc42T deactivation, 2) decreasing the rate constant of dissociation of the Cdc42T-GEF complex $k_{4b}$, 3) increasing the rate constant $k_{5a}$ for membrane binding of cytosolic Cdc42D and 4) increasing the rate constant $k_6$ for association of Cdc42T with cytosolic GEF to form the Cdc42T-GEF complex. We computed patch mobility in simulations where both $k_{2b}$ and $k_{5a}$ where modified, (Fig 5C and 5D) and where $k_6$ was increased (Fig 5E and 5F), because these changes resulted in polarization down to 15 GEF molecules. Interestingly, as the total GEF abundance was decreased, only increasing $k_6$ generated highly mobile patches (compare Fig 5D and 5F which are representative realizations from the points indicated by the red arrows in Fig 5C and 5E).

The rate constant $k_6$ governs the direct recruitment of cytosolic GEF to the patch through complex formation with Cdc42T. This reaction can occur rapidly, because diffusion in the cytosol is fast compared to diffusion in the membrane. Another potentially fast reaction not considered in the model is the recruitment of cytosolic Cdc42D directly to the patch. In cells, most inactive Cdc42 molecules are found in the cytosol bound to GDI proteins that hold them in their inactive state. In our original model, for a cytosolic Cdc42D molecule to be activated it must first be inserted in the membrane, a step implicitly representing dissociation from the GDI protein. Once at the membrane Cdc42D has to laterally diffuse to react with a GEF. However, prior work has suggested that GEFs may displace Rho-GTPases from their GDI proteins [34,35]. Based on this observation, we updated our model to include a reaction where Cdc42T-GEF recruits and activates cytosolic Cdc42D (Fig 5G). An analogous reaction was included in a model for polarization by Klunder et al. (2013) [33]. Using a rate constant comparable to the one employed by Klunder et al. and maintaining the original values of the other rate constants, we observed a highly mobile patch at low GEF abundances (Fig 5G and 5H). Our results suggest that direct recruitment of fast diffusing cytosolic polarity factors to the patch promotes mobility of the polarity cluster.

To gain further insight into mechanisms that generate a dynamic patch, we investigated several different patch properties. We first observed that the models shown in Fig 5C and 5G, which have substantially different patch mobility at low GEF abundance (replotted in Fig 6A on a log-log scale as "High mobility" and "Low mobility" models), have nearly identical amounts of active Cdc42 and Cdc42T-GEF complex, and the fluctuations in the abundance of these species were also similar between the two models (S4 Fig). However, the dwell time of

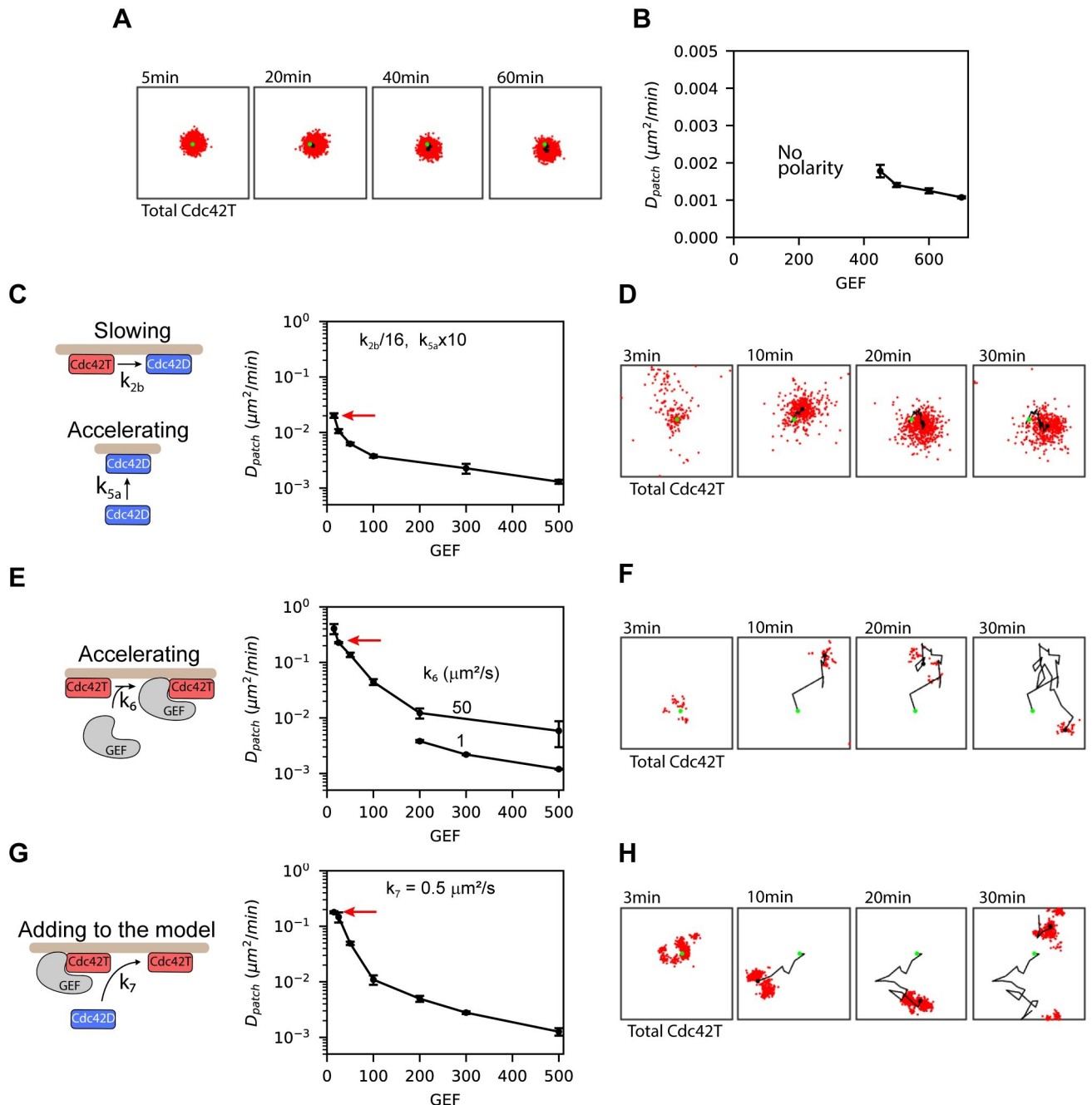

**Fig 5. Direct recruitment of polarity factors to the patch enables highly mobile clusters. (A)** Snapshots of the distribution of total active Cdc42 over a 1hr simulation. The black dot in each frame is the centroid of the polarity cluster, and the green dot is the centroid when the polarity cluster first formed. **(B)** Diffusivity of the centroid of the polarity patch ($D_{patch}$) as a function of the total amount of GEF molecules in the simulations. $D_{patch}$ was obtained from the mean squared displacement (MSD) of the patch centroid by fitting the equation $\text{MSD}(\Delta t_i) = 4D_{patch}\,\Delta t_i^{\beta}$ to the data, where $\Delta t_i$ is a particular time interval, and β reflects the degree of anomalous diffusion. **(C)** Patch diffusivities as a function of total GEF for simulations where $k_{2b}$ has been decreased by a factor of 1/16 and $k_{5a}$ has been increased by a factor of 10 relative to the parameters in Table 1. For these simulations β ≈ 1. **(D)** Snapshots from a representative simulation in **(C)** as indicated by the red arrow. **(E)** Patch diffusivities as a function of total GEF for simulations where $k_6$ has been increased to either 1 μm²/s or 50 μm²/s. With $k_6 = 1$ μm²/s polarization is lost when the number of GEFs is below 200. With $k_6 = 50$ μm²/s the simulations show polarization at even lower GEF amounts, in this case, β varied between 0.85 and 1. **(F)** Snapshots from a representative simulation in **(E)** as indicated by the red arrow. **(G)** Patch diffusivities as a function of total GEF after adding Reaction 7 to the model. β values were ≈ 0.85 for the two data points with highest mobilities, and close to 1 for the other points. **(H)** Snapshots from a representative simulation in **(G)** as indicated by the red arrow. Error bars for patch centroid diffusivities are standard errors from the least-squares fit use to compute $D_{patch}$.

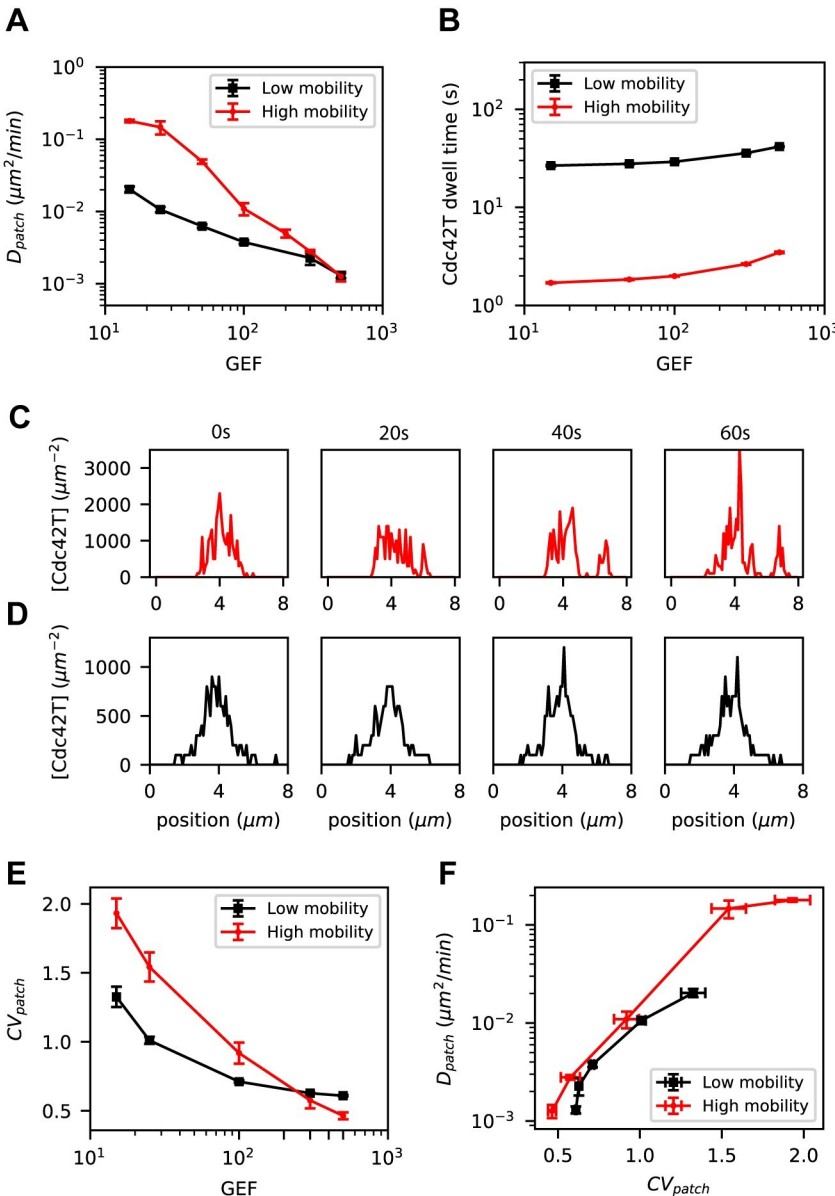

**Fig 6. Cluster mobility correlates with fast distribution fluctuations and short dwell times of Cdc42T.** Comparison of a "Low mobility" model (from Fig 5C) and a "High mobility" model (from Fig 5G). **(A)** Patch centroid diffusivities, **(B)** dwell time of Cdc42T at the cluster, **(C)** Snapshots of the lateral profile of the concentration of total Cdc42T molecules for the High mobility model and **(D)** Low mobility model. **(E)** Coefficient of variation of the distribution of total Cdc42T molecules, $CV_{patch}$ (see Methods), as a function of the number of total GEF molecules. **(F)** Patch centroid diffusivity as a function $CV_{patch}$ for the Low mobility and High mobility models. Error bars for patch centroid diffusivities are standard errors from the least-squared fit used to compute $D_{patch}$. For all other quantities, the error bars are the standard deviation from estimations in 5 independent simulations.

Cdc42T was shorter in the high mobility model, indicating a faster cycling of Cdc42 between the cluster and the cytosol as compared to the low mobility model (Fig 6B).

We also observed that patches with high mobility appeared to show larger spatial fluctuations in the distribution of active Cdc42 (Fig 6C), as compared to the low mobility case (Fig

6D). We quantified spatial fluctuations in the Cdc42T distribution by computing the coefficient of variation $CV_{patch}$ (see Methods) and observed that this quantity strongly correlates with $D_{patch}$ (Fig 6E).

These results suggest that patch mobility is correlated with spatial fluctuations in the Cdc42T distribution, rather than fluctuations in the abundance of Cdc42T or its GEF. Furthermore, low GEF abundance and rapid Cdc42 cycling between the cluster and the cytosol are required for high patch mobility.

**B.2. Potential mechanisms for regulating polarity cluster dynamics.** During mating, the polarity site transitions from being highly mobile to a more stable state. Therefore, we investigated mechanisms that would allow regulation of patch movement. We note that increasing $k_6$ or adding Reaction 7 in our original model enabled highly mobile clusters at low GEF abundances, but reversing such modifications eliminates polarization instead of stabilizing the patch (see for example Fig 5E, $k_6 = 1$ µm$^2$/s). We therefore systematically evaluated how all other reactions affect patch mobility in a model that included Reaction 7, which we refer to as the *updated model* (Figs 7A–7C, S5 and S6). We found that the following reactions robustly modulate cluster mobility: activation of Cdc42D$_m$ by GEF$_m$ ($k_{2a}$), activation of Cdc42D$_m$ by Cdc42T-GEF ($k_3$) and association of Cdc42T with GEF$_m$ to form the Cdc42T-GEF complex ($k_{4a}$). We note that these reactions all involve the association between two membrane-bound molecules.

To investigate the role of $k_{2a}$, $k_3$ and $k_{4a}$ in stabilizing the polarity site, we systematically varied the GEF abundance and one of these rates, while setting the other two equal to zero in the cases of $k_3$ and $k_{4a}$ or a very small value in the case of $k_{2a}$ (a minimal value is required to start Cdc42 activation) (Fig 7A–7C). We found that increasing $k_{4a}$ was most effective at reducing patch mobility as it stabilizes the patch at lower values of the rate constant and a wider range of GEF abundances compared to $k_{2a}$ and $k_3$. To understand why varying $k_{4a}$ produced a more dramatic effect on patch mobility, we quantified the amounts and dwell times of Cdc42T and GEF at the patch as a function of $k_{4a}$, $k_{2a}$ and $k_3$ (Fig 7D–7I). Increasing any of the three rates produced a modest increase in Cdc42, but only increasing $k_{4a}$ produced a substantial increase in GEF at the patch (Fig 7D–7F). On the other hand, increasing $k_{2a}$ and $k_3$ lengthened the dwell time only of Cdc42, while increasing $k_{4a}$ extended the dwell time only of GEF (Fig 7G–7I). These results suggest that $k_{2a}$ and $k_3$ stabilize the patch mainly by prolonging the residence time of Cdc42 at the patch, while $k_{4a}$ reduces cluster mobility by trapping GEF at the patch and increasing its abundance.

Because $k_{4a}$ had the largest effect on patch mobility, we studied the effects of varying this parameter using the original set of parameter values in the updated model. Setting $k_{4a}$ equal to zero resulted in an increase of more than an order of magnitude in patch mobility for low GEF abundances (Fig 8A). Increasing $k_{4a}$ with GEF = 100 resulted in an abrupt change in cluster mobility around a value of 0.003 µm$^2$/s (Fig 8B), in agreement with the results seen using the reduced values of $k_{2a}$ and $k_3$ in Fig 7C. After this sharp decrease, the mobility of the patch gradually increased with increasing $k_{4a}$ and appeared to plateau as the reaction becomes diffusion-limited.

## C. Validation through 3d particle-based simulations

So far, all our results have been obtained using 2D simulations with periodic boundary conditions and assumed the only difference between the membrane and cytosol is the rate at which molecules diffuse. However, real yeast cells are 3D objects with distinct membrane and cytosolic compartments. Therefore, to ensure our results remained valid in 3D, we used Smoldyn [22,23] to perform particle-based simulations on a sphere, translating 2D parameters to 3D as described in the Methods. Because varying $k_{4a}$ had the largest effect on patch mobility, we

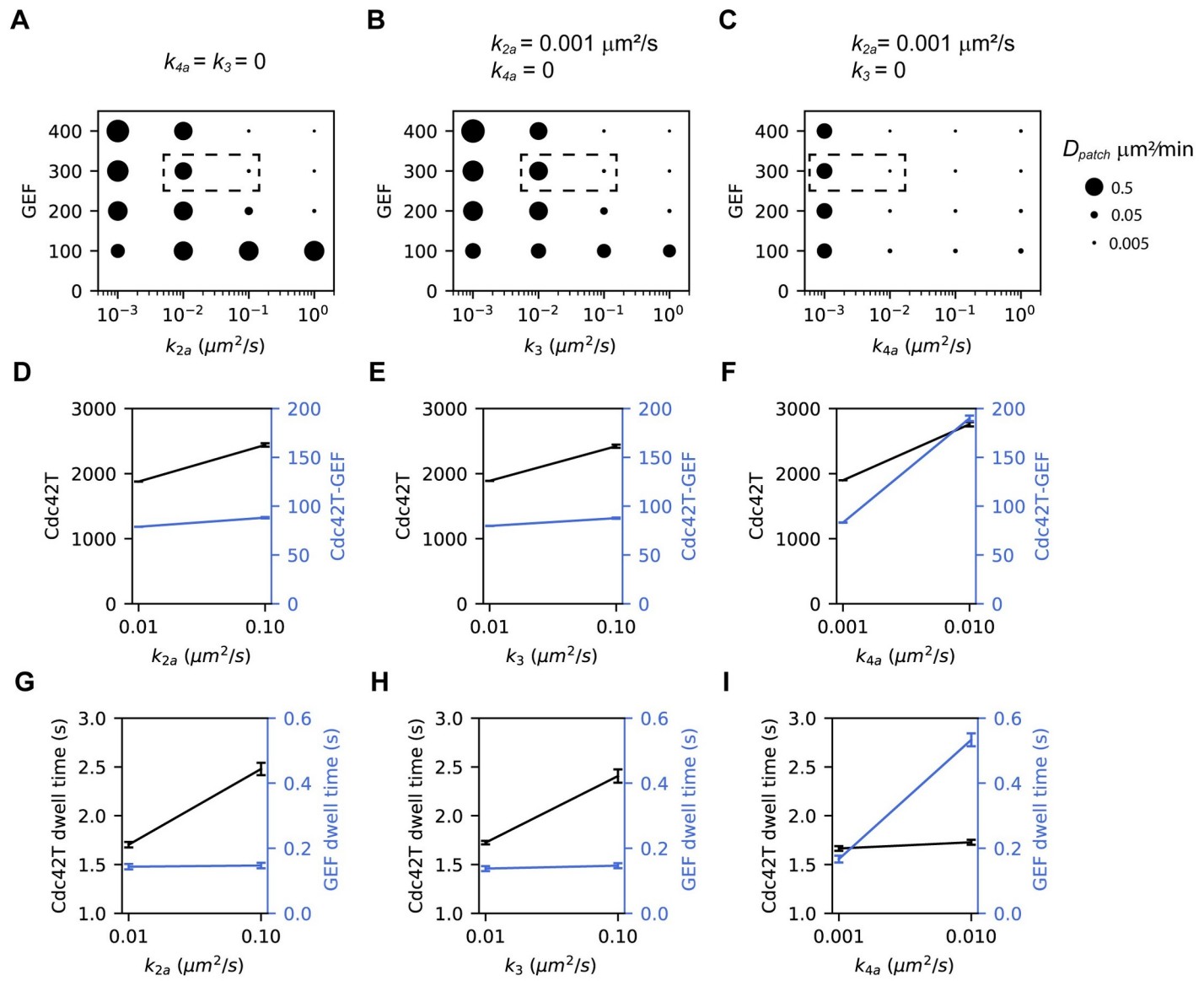

**Fig 7. Association reactions at the membrane stabilize clusters by trapping polarity factors at the patch.** Patch centroid diffusivity ($D_{patch}$) as a function of the total number of GEF molecules and $k_{2a}$ (**A**), $k_3$ (**B**), $k_{4a}$ (**C**). The size of the dots reflects the magnitude of $D_{patch}$ as indicated in the legend at the right. For each case, the value of the other two rate constants is shown above the panel. (**D-F**) The amounts of Cdc42 and GEF at the patch are quantified as the total Cdc42T (black) and Cdc42T-GEF (blue) respectively for the corresponding points enclosed by dashed boxes in (**A-C**). (**G-I**) Dwell times at the patch of Cdc42T (black) and GEF (blue) for the corresponding points enclosed by dashed boxes in (**A-C**) (see Methods for details). Error bars are the standard deviation from estimations in 10 independent simulations.

chose to validate these results. In agreement with our 2D simulations, low values of $k_{4a}$ produced a mobile polarity patch and patch mobility decreased rapidly with increasing values of this rate constant (Fig 9A). Quantifying patch movement as a function of $k_{4a}$ in the 3D simulations produced similar trends as results from the 2D spatial Gillespie simulations (Fig 9B). We note, however, that in the 3D particle-based simulations, the transition from a mobile to static patch appears to take place at a slightly higher values of $k_{4a}$ (between 0.005 and 0.01 $\mu m^2/s$), and for each value of $k_{4a}$, $D_{patch}$ is higher, in comparison with the 2D spatial Gillespie simulations. The good agreement between the full 3D and approximate 2D simulation results

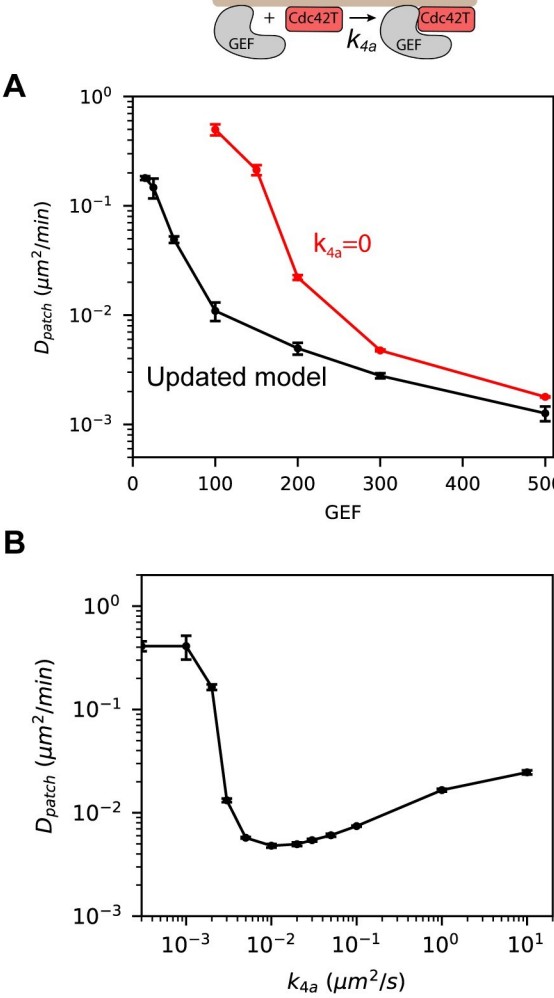

**Fig 8. Increasing the rate constant of Cdc42T-GEF$_m$ association ($k_{4a}$) induces an abrupt change in patch mobility.**
(**A**) Patch centroid diffusivity as a function of the number of GEF molecules for the model that includes Reaction 7 (Updated model, black) and the same model but setting $k_{4a} = 0$ (red). (**B**) Patch diffusivity as a function of $k_{4a}$ for simulations with GEF = 100. Error bars are standard errors from the least-squared fit used to compute $D_{patch}$.

validates the use of the more computationally efficient 2D simulations to investigate dynamics of the polarity patch. These results also provide further support for a mechanism for stabilizing the polarity patch by increasing the rate at which membrane-bound GEF and Cdc42T associate.

# Discussion

How cells relocate polarity clusters at the cell membrane during different tasks such as migration [36,37], growth [38,39], sporulation [40] and mating [5,41] is a fundamental question that has not been fully understood. By means of computational modeling, we investigated how molecular noise can be exploited to promote lateral mobility of polarity clusters and how cells can regulate cluster mobility. We focused on dynamic polarization observed during the early stages of mating in budding yeast. When haploid cells are presented with pheromone from

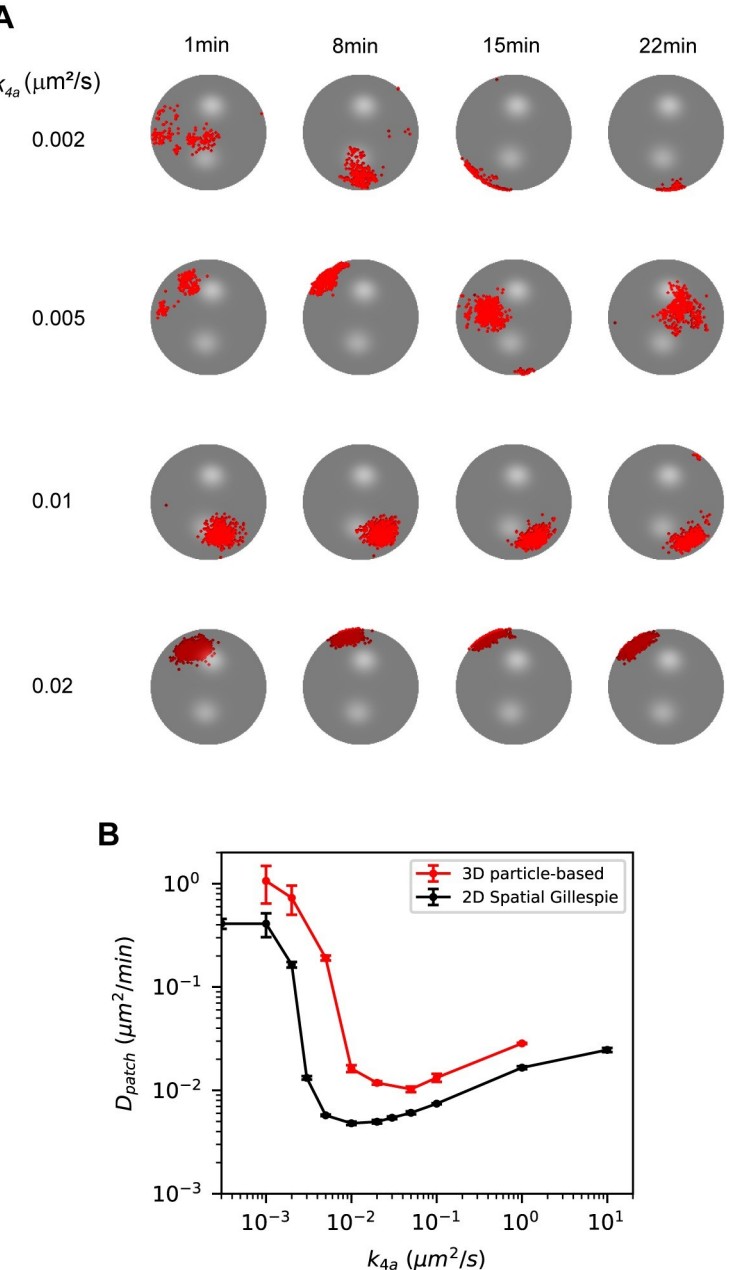

**Fig 9. 3D particle-based simulations recapitulate 2D spatial Gillespie simulations results from Fig 8. (A)**
Snapshots of 3D particle-based simulations for different values of $k_{4a}$. Cdc42T molecules are shown as red dots on a spherical surface representing the cell membrane. Rate constants are estimated from the ones used in Fig 8B as described in the Methods. Parameters are presented in Table 1. **(B)** Patch centroid diffusivity as a function of $k_{4a}$ for 3D particle-based simulations (red) with GEF = 100. For comparison we also show results from 2D spatial Gillespie simulations in Fig 8. Error bars are standard errors from the least-squared fit used to compute $D_{patch}$.

potential mating partners, the Rho-GTPase Cdc42 forms highly dynamic clusters that move across the membrane and stabilize in the direction of an adjacent cell of the opposite mating type.

## Efficient and accurate stochastic simulations

To investigate the effect of molecular-level fluctuations on cluster mobility, we performed stochastic simulations of the biochemical network of Cdc42 polarization in yeast. Stochastic effects are most accurately captured using particle-based (microscopic) simulations [16,22,23,42–44] or a convergent reaction-diffusion master equation [45]. However, the long-time scales associated with the movement of the polarity site make the use of such simulations computationally prohibitive to perform extensive investigations. We therefore used less accurate, but more computationally efficient simulations based on the spatial Gillespie method. Despite improvements on the accuracy of spatial Gillespie simulations [18–20], we found that simulations lost accuracy in the diffusion-limited regime at high molecular concentrations. To increase the accuracy of such simulations, we implemented concentration-dependent mesoscopic rate constants building on ideas from Yogurtcu et al [21]. An alternative approach that could also provide accurate and computationally efficient results is using hybrid microscopic-mesoscopic simulations [46–51], for example coupling a microscopic formulation of the cell membrane with a mesoscopic representation of the cytosol. Such methods, of course, imply higher implementation complexity.

## Highly dynamic clusters

Relocation of polarity clusters has commonly been explained via negative feedback mechanisms [52]. Negative feedback reactions can destabilize positive-feedback driven clusters resulting in travelling waves [36], or oscillations where the clusters disappear and reappear at different locations of the membrane [41,53]. Directed vesicle delivery is also a form of negative feedback that dilutes the polarity cluster and induces wandering motion [7,13,14]. Here, we document how biochemical noise can induce relocation of polarity clusters without an explicit negative feedback mechanism. Essential for noise-driven cluster motion are low abundances of limiting polarity factors and fast cycling of polarity factors between the cluster and the cytosol. Fast cluster-cytosol cycling can be promoted by reactions where polarity factors are directly recruited to the cluster.

## Regulation of cluster dynamics

During yeast mating, stabilization of highly mobile clusters has been attributed to increased cytosolic levels of the polarity factor Cdc24, the Cdc42 GEF, as pheromone induced MAPK activity triggers its nuclear export [5]. Besides increased GEF abundance, our study suggests that accelerating the kinetics of second-order reactions at the membrane involved in the activation of Cdc42 can stabilize highly dynamic clusters. Increasing the rate constants of such reactions seem to stabilize a mobile cluster by increasing the abundance and trapping polarity factors at the patch. Interestingly, modulating the rate constant of one such reaction, association of membrane-bound Cdc42 and GEF, produced a switch-like transition from a mobile to a stable polarity patch. The Cdc42/GEF interaction, therefore, is a likely target of pheromone induced signaling during regulation of polarity cluster dynamics. Interestingly, this interaction, which in yeast cells is bridged by the scaffold protein Bem1, is thought to be regulated in another context involving cell-cycle control [54].

Further evidence highlights the importance of biochemical events taking place at the cell membrane, and properties of the cell membrane itself in the regulation of polarity cluster dynamics. In fission yeast cells, which also display mobile patches in the early stages of mating, cluster dynamics is known to be under the control of a GAP (Cdc42 inactivator molecule) that localizes at the cell membrane [55]. During spore germination of fission yeast, initial uniform growth is associated with highly dynamic Cdc42 clusters. Upon rupture of the outer spore

wall, the clusters stabilize into a single cluster in the direction where rupture takes place, giving rise to directed growth [40]. Other studies have documented that membrane tension [56] and membrane curvature [57,58] can influence cluster stability. Additional mechanisms that may be used by cells to regulate biochemical events at the membrane and control cluster dynamics include crowding of signaling molecules [59,60], restricting the diffusion of molecules with cytoskeletal barriers [11,61,62] and confining molecules into high affinity subdomains [63].

In summary, our results demonstrate the power of using accurate and efficient mesoscopic simulations to inform more detailed, but computationally costly, particle-based simulations. Our studies also provided considerable insight into the mechanisms used by cells to harness random molecular behavior and regulate the dynamic properties of their polarity sites.

## Methods

### Spatial Gillespie simulations

In this coarse-grained approach, space is discretized into grid units, and the state of the system is given by the number of molecules of each species in each grid unit. The system evolves continuously in time according to the reaction-diffusion master equation, which is the spatial extension of the chemical master equation for well-mixed systems. We simulated individual realizations with the Next Subvolume method [17] which is an efficient implementation of the spatial version of the stochastic simulation algorithm [64]. We ran simulations on a square domain of size $L$ discretized with a Cartesian mesh with grid element size $h$. In the spatial Gillespie algorithm, diffusion is treated as a reaction that results in a molecule transitioning from its current location to a neighboring grid unit. If there are $n$ molecules of a given species in a particular grid unit, the propensity $k_{jump}$ for one of those molecules to transition to a neighboring grid unit is $n\,D/h^2$, where $D$ is the diffusion coefficient. In 2D, there are 4 neighbor cells and the total propensity of jump is $4\,n\,D/h^2$. The reaction propensities within a grid unit are estimated in the same way as the well-stirred Gillespie algorithm [64]. For example, the propensity of a second-order reaction for the association of the species A and B, is computed as $k_{meso}\,n_A\,n_B/h^2$ where $k_{meso}$ is the mesoscopic rate constant and $n_A$ and $n_B$ are the numbers of molecules A and B in the particular grid unit. The difference in the versions of the spatial Gillespie methods we use here are the different ways in which $k_{meso}$ is computed.

### Microscopic models for reacting particles

There are two common and related models for describing bimolecular reactions at the microscopic level, we refer to them as the Smoluchowski model [24], and the λ-ρ model [25,26]. Both consider diffusing molecules as particles undergoing Brownian motion, but they differ in how reactions are described. The Smoluchowski model represents reacting molecules as solid spheres that can react when separated exactly by a distance σ. In this model a molecule A will react with probability rate $k_{micro}[B]_\sigma$, where $[B]_\sigma$ is the concentration of its reactive partner B at the reactive distance σ. In the λ-ρ model, reacting molecules are considered point-particles that react with probability rate λ when they are separated by a distance equal or less than ρ.

Both formalisms provide expressions for the macroscopic rate constant $k$ in terms of the microscopic parameters for 3D systems [24,26]. In the reaction-limit where $4\pi\sigma D >> k_{micro}$ (Smoluchowski model) or $D >> \lambda\,\rho^2$ (λ-ρ model), where $D$ is the sum of the diffusion coefficients of A and B, $k = k_{micro}$ and $k = 4\pi\rho^3\lambda/3$ in the Smoluchowski and λ-ρ models respectively. In the opposite limit, when reactions take place immediately when the molecules meet (diffusion-limit), the corresponding equations are $k = 4\pi\sigma D$ and $k = 4\pi\rho D$.

In 2D there are no general closed form expressions relating macroscopic and microscopic rate constants. However, in the reaction-limit, which in 2D requires $D >> k_{micro}$ in the Smoluchowski model or $D >> \lambda\rho^2$ in the $\lambda$-$\rho$ model, macroscopic rate constants can be approximated as $k = k_{micro}$ and $k = \pi\rho^2\lambda$, respectively. For diffusion-limited reactions macroscopic rates are not well defined and depend on local concentrations [21].

Because it is easier to implement, we followed the $\lambda$-$\rho$ formalism and treated particle-based simulations based on this description as the ground truth. However, because the derivations of $k_h$ and $k_c$ make use of the Smoluchowski model, to compare particle-based and RDME simulations we take $\sigma = \rho$ and used an approximate relation between $\lambda$ and $k_{micro}$. To obtain this relation we formulated a microscopic rate equation for the probability rate that a molecule A reacts when there is a B molecule within a distance $\rho$:

$$\lambda = k_{micro}[B]_\rho, \tag{1}$$

and defining $[B]_\rho = 1/(\pi\rho^2)$ in 2D we obtain:

$$k_{micro} = \lambda\pi\rho^2. \tag{2}$$

## The scale-dependent mesoscopic rate constant $k_h$ in a 2D system

To account for the fact that, in the RDME setting, the encounter time of two molecules, and therefore, the reaction time, diverges as the grid element size decreases, Hellander et al. [19] derived a mesoscopic rate constant $k_h$ using the condition that the mean association time for two molecules diffusing in a specified domain in the RDME representation is equivalent to the exact result for a microscopic description following the Smoluchowski approach.

The mean association time $\tau_{micro}$ for two molecules diffusing on a domain with area $A$ in the microscopic formulation can be estimated from [18]. Assuming that one molecule diffuses on a circular domain with reflective boundary conditions and radius $R$ such that $\pi R^2 = A$, and the other is static at the center of the domain:

$$\tau_{micro} = \frac{\pi R^2}{k_{micro}}[1 + \alpha F(v)] \tag{3}$$

with

$$F(v) = \frac{\ln(1/v)}{(1 - v^2)^2} - \frac{3 - v^2}{4(1 - v^2)}, \tag{4}$$

$$v = \frac{\rho}{R}, \tag{5}$$

where $\rho$ is the reactive radius and the parameter $\alpha$ is defined as:

$$\alpha = \frac{k_{micro}}{2\pi D}, \tag{6}$$

where $D$ is the sum of the diffusion coefficients of the two molecules.

The mean association time in the mesoscopic formulation $\tau_{meso}$ is estimated for a square domain of length $L$ with square grid units of length $h$ [19] as:

$$\tau_{meso} = \frac{L^2}{2\pi D}\log\left(\frac{L}{h}\right) + \frac{0.1951 L^2}{4D} + \frac{L^2}{k_{meso}}. \tag{7}$$

$k_h$ is the mesoscopic rate constant that ensures that $\tau_{meso} = \tau_{micro}$ provided that $L^2 = \pi R^2$. This

leads to the following expression for $k_h$:

$$k_h = k_{micro} \left[ 1 + \frac{k_{micro}}{D} G \right]^{-1} \tag{8}$$

where

$$G = \frac{1}{2\pi} \log\left(\frac{h}{\sqrt{\pi}\rho}\right) - \frac{1}{4}\left(\frac{3}{2\pi} + 0.1951\right). \tag{9}$$

Note that $k_h$ can be computed only if:

$$1 + \frac{k_{micro}}{D} G > 0. \tag{10}$$

This implies a lower bound on $h$:

$$h > \sqrt{\pi} e^{\frac{3+2*0.1951\pi}{4} - \frac{2\pi D}{k_{micro}}} \rho. \tag{11}$$

When $h$ is below this bound, there is not $k_h$ for which the equality $\tau_{meso} = \tau_{micro}$ holds.

## Mesoscopic dissociation rate $k_h{}^d$

The mesoscopic dissociation rate constant is derived so that the steady state concentrations of a reversible second-order reaction of a two-molecules system are the same in the mesoscopic and microscopic formulations [20]. The steady state is characterized by the ratio of the average unbound time to the total time, which can be computed as the ratio of the mean rebinding time to the sum of the mean rebinding time and the mean dissociation time. Therefore, the condition to be satisfied is:

$$\frac{\tau_{meso}^{rebind}}{\tau_{meso}^{rebind} + \tau_{meso}^d} = \frac{\tau_{micro}^{rebind}}{\tau_{micro}^{rebind} + \tau_{micro}^d}. \tag{12}$$

The mean rebinding time in the mesoscopic simulation $\tau_{meso}{}^{rebind}$ was shown to be $L^2/k_{meso}$ and a good approximation for the mean rebinding time in the microscopic formulation $\tau_{micro}{}^{rebind}$ in the 2D disk domain is $L^2/k_{micro}$. The mean dissociation times in the mesoscopic ($\tau_{meso}{}^d$) and microscopic ($\tau_{micro}{}^d$) formulations are respectively $1/k_{meso}{}^d$ and $1/k_{micro}{}^d$. After replacing $k_{meso}$ and $k_{meso}{}^d$ by $k_h$ and $k_h{}^d$ the equilibrium condition then reduces to:

$$k_h^d = k_{micro}^d \frac{k_h}{k_{micro}}. \tag{13}$$

## Mesoscopic concentration-dependent rate constant $k_c$

The mesoscopic rate constant $k_c$ aims to provide accurate results in systems with high concentrations relative to the space discretization. In this case, several reacting molecules are often found together in a grid element. Therefore, reaction rates are dominated by the time it takes for molecules to react once they are in the same grid element and it is reasonable to ignore the time for molecules to diffuse into the same grid unit.

Let us consider a grid element containing a single A molecule and one or more of its reacting partner B at a concentration $[B]_h$. In this scenario we define $k_c$ via the mesoscopic rate equation:

$$k_c[B]_h = 1/\tau_{Rc}, \tag{14}$$

where $\tau_{Rc}$ is the mean reaction time. To account for the fact that at higher $[B]_h$ the mean-free-path before reaction is shorter (and therefore, so is the reaction time), $\tau_{Rc}$ is approximated as the mean association time of two molecules diffusing on a domain with area $A_c = h^2/n_B$ where $n_B$ is the number of B molecules in the grid element. The time $\tau_{Rc}$ is computed using Eq 3 with:

$$R = R_C = \sqrt{\frac{A_c}{\pi}}. \tag{15}$$

Taking $[B]_h = h^2/n_B = 1/A_c$ and replacing in Eq 14 we get:

$$k_c = k_{micro}\left[1 + \alpha F\left(\frac{\rho}{R_c}\right)\right]^{-1}, \tag{16}$$

where the function $F$ and the constant $\alpha$ are defined in Eqs 4 and 6.

For crowded situations where $R_c \leq \rho$, we set $k_c = k_{micro}$. Although technically there is not a lower bound on $h$, for realistic simulations $h$ should be greater than $\rho$.

In the more general case where there can be more than one A or B molecules in the grid element we get:

$$A_c = \frac{h^2}{\max(n_A, n_B)}, \tag{17}$$

where $\max(n_A, n_B)$ is the number of the most abundant reactant within the grid, and $k_c$ is computed from Eq 16 using:

$$R_C = \sqrt{\frac{h^2}{\pi\max(n_A, n_B)}}. \tag{18}$$

## Mesoscopic dissociation rate $k_c{}^d$

We estimate the mesoscopic dissociation rate $k_c{}^d$ in a similar way as described above for a two-molecule system (Eqs 12 and 13) under the assumption that the pair of molecules diffuse on a domain with area $A_c$:

$$k_c^d = k_{micro}^d \frac{k_c}{k_{micro}}. \tag{19}$$

## 2D particle-based simulations

The simulations performed to benchmark our methods (Fig 2) were carried out using our own custom written software. All particle-based polarity simulations (2D and 3D) were performed using Smoldyn [22,23]. In all particle-based simulations used the λ-ρ approach for second-order reactions. While this is not the default scheme in Smoldyn, which typically approximates the original formulation of Smoluchowski model where reacting molecules always react upon contact, the software also admits λ-ρ parameters as described below.

In these particle-based simulations space is continuous and time is discretized in intervals $\Delta t$. Molecules are considered point particles and their Brownian motion is simulated with the Euler-Maruyama method: If $x(t)$, $y(t)$ are the position coordinates of a given particle at time $t$ moving in a 2D domain, the position coordinates at time $t + \Delta t$ are calculated as:

$$x(t + \Delta t) = x(t) + Z_i\sqrt{2D_m\Delta t}, \tag{20}$$

$$y(t + \Delta t) = y(t) + Z_j\sqrt{2D_m\Delta t} \tag{21}$$

where $Z_i$, and $Z_j$ are independent random numbers drawn from a standard normal distribution, and $D_m$ is the diffusion coefficient. Every time step, the new positions of all the particles are calculated.

Bimolecular reactions occur with probability P = 1-exp(-λ Δ*t*) during a time interval Δ*t*, when two reactants are within a distance ρ (the reaction radius). This probability is approximated as P ≈ λ Δ*t* for small Δ*t*. For simulations with Smoldyn we input to the software the probability P.

First-order reactions in our custom made simulations (Fig 2), occur during a time interval Δ*t* with probability $P_i$ = 1-exp(-$k_i$ Δ*t*), which can be approximated as $P_i$ = $k_i$ Δ*t* for small Δ*t*, where $k_i$ is the reaction rate constant. For simulations performed with Smoldyn we only input the rate constant.

In our custom-made simulations (Fig 2), when a dissociation event for two molecules in a complex occurs, one molecule is set at the position previously occupied by the complex and the second is placed at distance ρ + ε apart from the first. ε is a small number just to ensure reactants will not be within a reactive distance the next simulation time step, and the orientation of the second molecule is chosen randomly from a uniform distribution. For simulations with Smoldyn, dissociation reactions are handled with the software default routines, but we specify the separation distance after dissociation as ρ + ε.

Simulations of simple reversible and irreversible reactions (Fig 2) were performed with Δ*t* ≈ (0.1ρ)$^2$/(4$D_{tot}$) with $D_{tot}$ = 2$D_m$. This ensures a root mean squared displacement < 0.1ρ over a simulation time-step resulting in accurate results at the expense of computational resources. The parameters of such simulations are given in the captions of the corresponding figures.

In the polarity simulations we used Δ*t* = (ρ)$^2$/(4$D_{cyto}$) after observing similar results with preliminary simulations using smaller time steps. In this case, the root mean squared displacement over a simulation time-step for cytosolic components is ρ, and it is less than ρ for membrane components since $D_{cyto} >> D_{memb}$.

The parameters for the polarity establishment model are given in Table 1. We present the reaction parameters as microscopic rate constants $k_{micro}$. For second-order reactions, the input to Smoldyn is the reaction probability P = λ Δ*t*, and λ is related to $k_{micro}$ as λ = $k_{micro}$/πρ$^2$ according to Eq 2.

## Reactions of the polarization model

$GEF_c \rightarrow GEF_m$ R-1a
$GEF_m \rightarrow GEF_c$ R-1b
$GEF_m + Cdc42D_m \rightarrow GEF_m + Cdc42T$ R-2a
$Cdc42T \rightarrow Cdc42D_m$ R-2b
$Cdc42T\text{-}GEF + Cdc42D_m \rightarrow Cdc42T\text{-}GEF + Cdc42T$ R-3
$Cdc42T + GEF_m \rightarrow Cdc42T\text{-}GEF$ R-4a
$Cdc42T\text{-}GEF \rightarrow Cdc42T + GEF_m$ R-4b
$Cdc42D_c \rightarrow Cdc42D_m$ R-5a
$Cdc42D_m \rightarrow Cdc42D_c$ R-5b
$Cdc42T + GEF_c \rightarrow Cdc42T\text{-}GEF$ R-6
$Cdc42T\text{-}GEF + Cdc42D_c \rightarrow Cdc42T\text{-}GEF + Cdc42T$ R-7

In the above reactions, $GEF_c$ and $GEF_m$ are cytosolic and membrane bound GEF, respectively. $Cdc42D_c$ and $Cdc42D_m$ are cytosolic and membrane-bound inactive Cdc42 (Cdc42-GDP), respectively. Cdc42T is membrane-bound active Cdc42 (Cdc42-GTP). Cdc42T-GEF is the membrane-bound complex of Cdc42T and GEF.

## Rate constants for the 2D stochastic polarization model

The rate constants for the 2D model in Fig 3A, presented in Table 1, were adapted from [13] which is a modified version of the model in [29]. That model is based on deterministic reaction-diffusion equations and is therefore a macroscopic representation. On the other hand, our stochastic simulations are parameterized with microscopic rate constants. For first-order and second-order reaction-limited reactions, the macroscopic rate constants from the model in [13] can be used directly in our simulations. However, for 2D diffusion-influenced reactions the conversion is more complicated [16]. For the purposes of this work, we used the macroscopic rate constants in [13] as a first approximation for the microscopic parameters and explored the effects of varying the rate constants over several orders of magnitude.

While the cell cytosol is a 3D compartment, in our 2D simulations the cytosol and the membrane are juxtaposed two-dimensional domains. This is a computationally efficient representation that neglects cytosolic gradients perpendicular to the membrane since, diffusion at the cytosol is fast compared to the timescale of reactions. To obtain equivalent rate constants for this purely 2D system we scaled cytosolic concentrations in the rate equations in [13] as:

$$[C_c]^{2D} = \frac{V_c}{A_m}[C_c]^{3D},$$ (22)

where $[C_c]^{3D}$ is the molar concentration of the cytosolic component $C_c$ in the original equations, $[C_c]^{2D}$ is the concentration of $C_c$ in the 2D cytosol, $V_c$ is the volume of the cell cytosol and $A_m$ is the membrane area.

In [13], concentrations at the membrane are expressed in molar units assuming that the membrane was a volumetric compartment with thickness $\Delta z$. We therefore also scaled concentrations of species at the membrane as:

$$[C_m]^{2D} = \Delta z[C_m]^{3D},$$ (23)

where $[C_m]^{2D}$ is the concentration of the membrane-bound species $C_m$ in units of mass/area, and $[C_m]^{3D}$ is the molar concentration of $C_m$. From the scaled reaction-diffusion equations we obtain the scaled 2D reaction rate constants $k^{2D}$ in terms of the rate constants $k^{3D}$ in [13]:

- Rate constants for first-order reactions that involve a transition from the cytosol to the membrane (R-1a and R-5a) are scaled as:

$$k^{2D} = k^{3D}\frac{\Delta z A_m}{V_c} = k^{3D}\eta.$$ (24)

- Second-order rate constants for reactions taking place at the membrane (R-2a, R-3, R-4a) are scaled as:

$$k^{2D} = k^{3D}\frac{1}{\Delta z}.$$ (25)

- Second-order rate constants for reactions in which a cytosolic species reacts with a membrane-bound species (R-6, R-7) are scaled as:

$$k^{2D} = k^{3D}\frac{A_m}{V_c}.$$ (26)

- The rate constants for first-order reactions in which the reactant and the product are bound to the membrane (R-2b and R-4b) are unchanged.

- Reactions 1b and 5b are the reverse of reactions 1a and 5a respectively, and to obtain the 2D rate constants it is necessary to multiply the corresponding volumetric rate constants by a factor of $1/\eta$. However, that factor cancels out with a factor of $\eta$ present in the reaction-diffusion equations used in [13,29], which takes into account the difference between the cytosol volume and the effective volume of the membrane. Therefore, $k_{1b}$ and $k_{5b}$ are unchanged.

The resulting 2D rate constants are presented in Table 1 using molecules as the unit of mass and μm as the unit of length.

We note one significant difference between the model parameters used here, and those used by McClure et al. We used a smaller number of total Cdc42 molecules based on the work of Watson et al. [65]. For our particle-based simulations, we used a smaller reaction radius $\rho$ than in our previous publication [16] to make this value closer to the size of the reacting proteins. This modification reduced the rate of Cdc42 activation, affecting polarization. To compensate for this effect, we increased the rate constants of Reactions 5a, 5b and 6 by a factor of 10.

### 3D particle-based simulations

3D simulations were performed using Smoldyn [22,23]. Membrane-bound species diffuse on the surface of a sphere with radius R and cytosolic components diffuse within the interior of the sphere. The parameter values used in the particle-based 3D simulations are given in Table 1. They correspond with the 2D parameters except when they have been converted to account for differences in the system dimensionality as described below.

The reaction rate constants were obtained from the 2D model. Rate constants for reactions that take place exclusively in the membrane do not need to be modified. These are Reactions 1b, 2a, 2b, 3, 4a, 4b and 5b.

The rate constants for reactions in which a cytosolic species binds the membrane or a membrane-bound molecule (Reactions 1a, 5a, 6, and 7) are estimated from the 2D model using the scaling introduced in the Methods subsection "Rate constants for the 2D stochastic polarization model" ignoring the factor $\Delta z$. With that scaling these 3D rate constants are obtained by multiplying the corresponding 2D rate constants by the factor $V_c/A_m$.

Although we report the microscopic rate constant for all reactions, we parameterized second-order reactions within Smoldyn, providing the probability of reaction P during a timestep $\Delta t$ when the separation of reactants is within $\rho$, approximated as $P = \lambda \Delta t$, for small $\lambda \Delta t$. For reactions where both reactants are at the membrane, the relation between $\lambda$ and $k_{micro}$ is the same as in the 2D simulations (Eq 2).

For second-order reactions where a cytosolic molecule reacts with a membrane-bound molecule and the product is at the membrane (Reactions 6, 7) $\lambda$ is expressed in terms of $k_{micro}$ as:

$$\lambda = k_{micro} \frac{3}{2\pi\rho^3}. \tag{27}$$

This follows from an expression equivalent to Eq 1 for the probability rate that a membrane bound molecule $A^m$ reacts with a cytosolic molecule $B^c$ located within a distance $\rho$:

$$\lambda = k_{micro}[B^c]_\rho, \tag{28}$$

where $[B^c]_\rho = 3/(2\pi\rho^3)$ is the concentration of a molecule in a half sphere of radius $\rho$.

## Quantification of clustering with $H(r)$

Before introducing the function $H(r)$ let us consider some typical metrics to quantify molecules aggregation. We first introduce the pair-wise molecule density $n(r)$:

$$n(r) = \frac{\sum_{i=1}^{N} m_i(r)}{N},$$

(29)

where $m_i(r) \, \Delta r$ is the number of molecules between distance $r$ and $r + \Delta r$ from molecule $i$ and $N$ is the total number of molecules. In a 2D domain with area $A$, if molecules are uniformly distributed, $n(r)$ should converge to $n_{unif}(r) = 2 \pi r (N-1) / A$. The pair-wise correlation function $g(r)$, commonly used in physics, can be computed as $g(r) = n(r)/n_{unif}(r)$, so that $g(r) = 1$ for a uniform distribution of molecules. Alternatively a distribution of pair-wise distances $P(r)$ can be obtained normalizing $n(r)$ by the $N$ -1 comparisons involved in the calculation of each $m_i(r)$ [31],

$$P(r) = \frac{n(r)}{N-1} = \frac{\sum_{i=1}^{N} m_i(r)}{N(N-1)}.$$

(30)

For a uniform distribution of molecules $P_{\mathrm{unif}}(r) = 2 \pi r / A$.

The $H(r)$ function quantifies the difference between the cumulative distribution function $\int_0^r P(r')dr'$ (sometimes referred to as the Ripley's K function) from the system of interest and the case with uniformly distributed molecules. In 2D, $H(r)$ is defined as:

$$H(r) = \sqrt{\frac{A}{\pi} \int_0^r P(r')dr'} - r.$$

(31)

With this definition $H(r) = 0$ if the molecule distribution has no structure (uniformly distributed) since $\int_0^r P_{unif}(r')dr' = \frac{\pi r^2}{A}$. Also, $H(r) < 0$ indicates dispersion and $H(r) > 0$ indicates clustering or aggregation. Furthermore, the value of $r$ where $H(r)$ is maximum provides an estimate of the cluster length-scale [30].

## Mean squared displacement (MSD) and effective polarity cluster diffusivity ($D_{patch}$)

After a polarity cluster has formed, the distribution of active Cdc42 is translated to the center of the domain to reduce border effects, and the centroid of the distribution is recorded every 1min. The centroid of the patch is calculated accounting for the toroidal geometry of the domain resulting from the periodic boundary conditions. The mean squared displacement (MSD) for a particular time interval $\Delta t_i$ is computed from all centroid trajectories over time intervals of length $\Delta t_i$ from multiple simulations. We discarded centroid jumps over the domain boundary by breaking trajectories containing jumps larger than a maximum jump $max_{jump}$ into sub-trajectories containing only jumps smaller than $max_{jump}$. We empirically set $max_{jump} = 6\mu m$ from visual inspection of centroid trajectories. Each MSD curve is obtained from 50 simulations of 1hr each except for S6 Fig where we used 5 simulations. The effective diffusivity of the polarity cluster ($D_{patch}$) can be obtained by fitting the equation $MSD(\Delta t_i) = 4D_{patch} \Delta t_i^{\beta}$ to the MSD data, where $\Delta t_i$ is a particular time interval, and a reflects the degree of anomalous diffusion. In practice we took logarithms to the data and fit the equation $\log(MSD(\Delta t_i)) = \log(4D_{patch}) + \beta \log(\Delta t_i)$ using only the first data points that showed a linear behavior in a log-log plot. The MSD during the smallest interval computed (1min) reflects rapid variations in the position of the centroid within the polarity cluster and do not contribute to the long scale displacement of the distribution. We therefore subtracted MSD(1min) to all MSD

data before estimating $D_{patch}$. MSDs in the 3D particle-based simulations were computed from geodesic displacements of the cluster on the spherical surface.

### Coefficient of variation of the distribution of active Cdc42T: $CV_{patch}$

$CV_{patch}$ was computed as a weighted average over space of the local coefficient of variation over time of the amount of Cdc42T. The average over space is weighted by the mean local abundance of Cdc42T:

$$CV_{patch} = \frac{\sum_j^A CV_j^T \langle Cdc42T_j \rangle^T}{\sum_j^A \langle Cdc42T_j \rangle^T}. \tag{32}$$

Here $CV_j^T$ is the mean divided by the standard deviation over a time interval T of the amount of Cdc42T at location $j$. $<Cdc42T_j>^T$ is the average amount of Cdc42T at location $j$ over a period of time $T$. The space average is computed over the whole simulation domain (A) as Cdc42T is mainly located at the polarity site. The time averages are computed over a short period $T = 1$ min to ensure the mean distribution of Cdc42T does not relocate significantly. One estimation of $CV_{patch}$ is obtained from a single simulation that has reached steady state with samples taken every second to compute time averages. In Fig 6E and 6F we plotted the mean and standard deviation (error bars) from 5 independent measurements of $CV_{patch}$.

### Dwell times at the patch

To compute the dwell time of Cdc42 at the patch, we introduced in the model additional tagged Cdc42 species (Cdc42$_m$$^{tagged}$, Cdc42T$^{tagged}$, Cdc42T$^{tagged}$-GEF) that have the same behavior as the untagged versions, except that if Cdc42$_m$$^{tagged}$ jumps from the membrane to the cytosol it converts into untagged Cdc42D$_c$. Simulations are initialized with no tagged species, and are run until the distribution of polarity factors reaches steady state. At this point, Cdc42$_m$, Cdc42T, Cdc42T-GEF are converted into the tagged versions in a region surrounding the polarity patch and we record the decay in the amount of the tagged molecules. The same idea is used to estimate the dwell time at the patch of GEF (introducing GEF$_m$$^{tagged}$ and Cdc42T-GEF$^{tagged}$). For Cdc42, we ignored the initial rapid decay coming from membrane detachment of inactive Cdc42. The dwell time at the patch is obtained by fitting an exponential decay function to the data. We reported the mean and standard deviation (error bars) from 10 or 30 independent measurements of the dwell time.

## Supporting information

**S1 Fig. Spatial Gillespie simulations of the reactions A+B → C and A+B ↔ C using the mesoscopic rates $k_h$ and $k_c$ benchmarked against particle-based results for different domain sizes.** We present the mean total number of species A as a function of time. The domain sizes are $L = 0.4$ μm (**A**, **B**), 0.8 μm (**C**, **D**), 1.6 μm (**E**, **F**), 3.2 μm (**G**, **H**) and 6.4 μm (**I**, **K**). Initial molecule abundances are A = B = 2000, C = 0. In all the simulations, the degree of diffusion control is $\lambda \pi \rho^2 / D_{tot} = 50$, with $D_{tot} = 2D$ and $D = 0.0025$μm$^2$/s, $\rho = 0.005$ μm and $\lambda = 3183.1$/s, with $D_{tot} = 2D$ and $D = 0.0025$μm$^2$/s. $h = 5\rho$ and the number of grid elements is $N^2 = (L/h)^2$. For the reversible reaction (**B**, **D**, **F**, **H**, **K**), the microscopic dissociation rate constant $k^d_{micro}$ is 10/s.
(TIF)

**S2 Fig. Fluctuations around the mean for spatial Gillespie simulations of the reactions A +B → C and A+B ↔ C using the mesoscopic rates $k_h$ and $k_c$ are compared particle-based**

**results.** The mean as a function of time is shown in Fig 2 of the main text. We present the standard deviation of total number of species A as a function of time. **(A-D)** show spatial Gillespie simulations using the mesoscopic rate $k_h$ with initial low abundance of reactants in **(A,B)** (total A = total B = 5, total C = 0 at t = 0) and initial high abundance (total A = total B = 5000, total C = 0 at t = 0) in **(C-D)**. **(E-H)** show corresponding simulations to **(A-D)** but using the mesoscopic rate $k_c$. In all the simulations, the degree of diffusion control is $\lambda\pi\rho^2/D_{tot} = 50$, with $D_{tot} = 2D$ and $D = 0.0025\mu m^2/s$, $\rho = 0.005$ μm and $\lambda = 3183.1/s$. The size of the domain is $L = 1\mu m$. For the reversible reaction, the microscopic dissociation rate constant $k^d_{micro}$ is 1/s in panels **(B, F)**, and 10/s in panels **(D, H)**.
(TIF)

**S3 Fig. Plots of H($r = 1.1$μm) (left panels) and total active Cdc42 (right panels) as a function of GEF molecules for different values of various model parameters ($k_{1a}$, $k_{1b}$, $k_{2a}$, $k_{2b}$, $k_3$, $k_{4a}$, $k_{4b}$, $k_{5a}$, $k_{5b}$, $k_6$).** In each panel the rate constant in the title of the figure is varied as indicated in the legend. For each parameter set, the simulations were initialized with an unpolarized random distribution with all GEF and Cdc42 in the cytosol and 10 min were simulated to provide enough time for the system to polarize. Mean and standard deviation (error bars) were calculated sampling every 30s for the last 5min of each simulation with data from 3 independent simulations.
(TIF)

**S4 Fig. Comparison of different metrics as a function of total GEF molecules for a "Low mobility" model (from Fig 5C) a "High mobility" model (from Fig 5G). (A)** Mean number of total Cdc42T, **(B)** coefficient of variation (*CV*) of total Cdc42T, **(C)** mean number of Cdc42T-GEF, **(D)** *CV* of Cdc42T-GEF.
(TIF)

**S5 Fig. The rate constant $k_3$ affects patch mobility when $k_{4a} = 0$. (A)** Effective diffusivity of the patch ($D_{patch}$) as a function of total available GEF in the updated model (including Reaction 7) for $k_3 = 0$ and $k_3 = 1$ μm²/s. **(B)** Similar to **(A)** except with $k_3 = 0$ and $k_3 = 1$ μm²/s keeping $k_{4a} = 0$. **(C)** Effective diffusivity of the patch ($D_{patch}$) as $k_3$ is varied with 300 GEF molecules. Error bars are standard errors from the least-squared fit used to compute $D_{patch}$.
(TIF)

**S6 Fig. Effective diffusivity of the polarity patch as a function of different parameters (indicated in the x-axis).** The panels on the left are for $k_{4a} = 2$ μm²/s and for the ones on the right $k_{4a} = 0$. Simulations were run with different GEF abundances as indicated. Missing points in each panel correspond to simulations that did not show robust polarization. Each data point was obtained from 5 simulations of 3600s each as described in the Methods. Error bars are standard errors from the least-squared fit used to compute $D_{patch}$.
(TIF)

**S1 Text. Descriptions for S3, S5 and S6 Figs.**
(DOCX)

## Acknowledgments

We thank all members of the Elston lab for helpful discussion throughout the project.

## Author Contributions

**Conceptualization:** Samuel A. Ramirez, Michael Pablo, Daniel J. Lew, Timothy C. Elston.

**Data curation:** Samuel A. Ramirez.

**Formal analysis:** Samuel A. Ramirez, Michael Pablo.

**Funding acquisition:** Daniel J. Lew, Timothy C. Elston.

**Investigation:** Samuel A. Ramirez, Michael Pablo, Sean Burk.

**Methodology:** Samuel A. Ramirez, Michael Pablo, Sean Burk.

**Writing – original draft:** Samuel A. Ramirez, Timothy C. Elston.

**Writing – review & editing:** Samuel A. Ramirez, Michael Pablo, Daniel J. Lew, Timothy C. Elston.

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
