## [Decision Letter · Decision Letter 0]

22 Mar 2021

Dear Dr. Ramirez,

Thank you very much for submitting your manuscript "A novel stochastic simulation approach enables exploration of mechanisms for regulating polarity site movement" for consideration at PLOS Computational Biology.

As with all papers reviewed by the journal, your manuscript was reviewed by members of the editorial board and by several independent reviewers. In light of the reviews (below this email), we would like to invite the resubmission of a significantly-revised version that takes into account the reviewers' comments.

We cannot make any decision about publication until we have seen the revised manuscript and your response to the reviewers' comments. Your revised manuscript is also likely to be sent to reviewers for further evaluation.

Sincerely,

Attila Csikász-Nagy

Associate Editor

PLOS Computational Biology

Mark Alber

Deputy Editor

PLOS Computational Biology

Reviewer's Responses to Questions

**Comments to the Authors:**

Reviewer #1: In this work, the authors explored the development of yeast cell polarization to an external pheromone gradient, investigating how the polarity factors formed clusters and then subsequently repositioned themselves before eventually forming a stable polarity site on the side of the pheromone source. This is entirely computational work, which builds on a model that the same group developed previously (McClure et al., Dev. Cell. 2015). The model is computationally intensive, so the authors developed new simulation methods that enable efficient simulation over an hour of real time; these new methods are a significant advance over existing ones. Intriguingly, they find that polarity cluster relocation does not necessarily rely on negative feedback, as had been posited previously, but can arise solely from stochastic noise.

Overall, this is very good work and appropriate for publication in PLOS Computational Biology. I have no concerns about any of the work presented, but I do have several suggestions for improvements, listed below. Note that I refer to several papers below, including from my own work, but am not requesting that the authors cite them; they may cite these papers if they wish but there is no pressure to do so.

Major issues

I tried to download the code used in this work from https://github.com/samuramirez/stochastic-exploratory-polarization, as the manuscript describes, but the repository is empty. This clearly needs to be fixed.

The reaction rate terminology was somewhat confusing. To most biochemists, including most modelers, there is just some given reaction rate constant for a given reaction. It was initially unclear how this rate constant related to the microscopic rate, the mesoscopic rate, the scale-dependent mesoscopic rate, and the concentration-dependent mesoscopic rate. Are some of these synonyms for each other? Also, is there any physical meaning to the mesoscopic rate constants, or are they essentially just simulation parameters that cause the simulation to agree accurately with the mean association times?

People who are familiar with Smoldyn know that its default simulation algorithm follows the Smoluchowski scheme, in which reactants only occur when two reactants are within a "binding radius" of each other and, if so, then react with probability 1. If the reaction is reversible, then Smoldyn separates the dissociation products by an "unbinding radius" that is typically larger than the binding radius. Also, Smoldyn has not been calibrated for quantitatively accurate reactions in 2D (see Johnson et al., BioRxiv 185595, 2020). However, this manuscript describes Smoldyn being used with Erban and Chapman's lambda-rho algorithm (Erban and Chapman, Phys. Biol. 6:046001, 2009) and having accurate 2D reactions, both of which are certainly possible with Smoldyn but not standard. I suggest checking what was really done and then clarifying this situation in the manuscript.

On the same topic, the text is misleading in the description of particle-based simulation in lines 805-813. I don't believe that the lambda-rho model uses the linear approximation for the probability. Also, for first order reactions, Smoldyn does not use the linear approximation for the reaction probability but instead uses the exact equation. Finally, for unbinding, Smoldyn's default behavior is to call the location of the complex the "reaction location" and to then separate both dissociation products away from that location in opposite directions, using a random orientation, but with total separation equal to the unbinding radius (which is larger than the binding radius). It's possible that the authors set Smoldyn up to use a different method here but this should be checked.

Particle-based simulations are being treated here as the ground truth against which the other simulation methods are compared. Is there independent evidence that the particle-based simulations are actually correct in these situations?

I suggest a little discussion about 2D reaction rate constants. They make good sense in PDE or RDME models but become a little less well defined in microscopic models. Are they actually meaningful in these contexts, and how should they be interpreted?

I was impressed by the similarity between the particle-based and lattice-based results presented in Figure 3. My experience is that it's relatively easy to get reaction rates to correspond, but it's harder to get spatial patterns to agree, and it can also be challenging to avoid lattice artifacts when using RDME-based models. See Andrews and Arkin, Current Biology, 16:R526, 2006.

Toward the end of section A.4, I appreciated the discussion of k_h vs. k_c. However, it wasn't clear what the conclusions were about which approach yields more accurate results, either in general or for specific situations, and how accurate those results are.

In section B.1, the authors explored the consequences of different GEF abundances. However, they didn't describe the current best guesses, or evidence, from the experimental literature, which would be useful to know.

In the discussion section on "Highly dynamic clusters", the text mentions that the model does not include an explicit negative feedback mechanism. Is there nevertheless some sort of implicit negative feedback in the model?

Minor issues

Line 142 - Would read better with "... accurately represent... "

Lines 144-147 - The reference to "scale-dependent rates" needs some explanation because most readers won't know what they are, at least at this point in the text.

Line 256 - Missing word "to".

Line 437 - Should replace "less" with "fewer".

Lines 815-817 - Do all membrane-bound molecules have the same diffusion coefficient? Also, it's worth pointing out that the rms displacement of a membrane-bound molecule is 0.1*rho, which is a very small step size (typically leading to slow simulations and high accuracy).

Line 824 - Is there a reference for this equation?

Line 828 - The rho exponent didn't print correctly.

Lines 850-851 - Was the previous model deterministic or stochastic?

Line 866 - The conversion between 2D and 3D certainly looks reasonable and yields the correct units, but is there a more rigorous justification for why it's correct?

Line 872 - I found the Th variable name to be confusing, especially since h is already a defined variable. How about delta-z, or something else that doesn't look like a product?

Line 941 - I wasn't familiar with the H(r) function or the Ripley's K function previously, and suspect that other readers might not know about them either, so I suggest slightly more explanation for them. Are they essentially just the integral of the radial distribution function, with a baseline offset?

For Table 1, I think it's worth pointing out that the 2D and 3D parameters are identical, except that a few cases required conversion due to dimensionality issues.

Reviewed by Steve Andrews

Reviewer #2: In this paper the authors implement a stochastic simulation method that accounts for fluctuations due to molecular noise. The proposed mesoscopic method allows many particles to occupy a lattice grid, within which they react with a concentration-dependent rate. The method is then applied to the Cdc42 polarity patch movement, which exhibits diffusive mobility due to these molecular fluctuations. Overall, the work is well motivated, the work performed is extensive and results would be interesting to many researchers. However, the paper was difficult for me to read. I used most of my effort to understand the main assumptions of the proposed simulation method (which as I state below are not so clear to me). This subsequent detailed parameter-dependence of a complex polarization model, with multiple panels (e.g. Fig. 6 and 7) was then hard to follow and evaluate in the context of the new simulation method as well interpret the new results for the Cdc42 system.

Main comments:

1) The new method could be better placed in the context of extensive prior work on simulations of reacting systems. Terms such as “high/low densities” and “diffusion-limited” are used but these terms are not given very precise mathematical definitions to identify the different regimes in the system. So it’s not clear to me if the cases of Fig. 2 are sufficient to validate the new method.

2) I think that the main assumption of the mesoscopic method, Eq. (14) needs to be clearly justified: is it rigorous and of not, what is the approximation introduced? This equation describes the reaction rate of molecules within the same grid. I see that in the reaction-controlled limit, the alpha*F term should be much smaller than one, and the reaction rate would obey law of mass action within the grid, as expected. When reactions start to be influenced by diffusion, then the alpha*F term that describes the resulting decrease in the reaction rate would start to become important. However, in this limit one would generally have a many-body reacting system within the grid, effectively described by time-dependent rate constants. Why is it ok to use a functional form for F that comes from a 2-particle calculation, over the grid diffusion time? Are any of the simulations in Fig. 2 and later in the limit where alpha*F is larger than one for it to matter?

3) I suggest that the authors simplify Figs. 5-9 to show the main conclusions with respect to mesoscopic method and any results that are important for the biology of the system. The analysis of the parameter dependence of the model is good, but I think it the readability will be greatly improved if the less important aspects are moved to an Appendix or Supplementary material.

More minor comments:

4) Why are the simulations of Fig. 2A,B,E,F shown only at small times compared to the equilibration time?

5) Why is the new method showing deviations that decrease with grid element size in Fig. 2F (line 297)?

6) How are the model parameters of Table 1 used in the different methods? Some comments in the table may help. For example, is the 3D model Smoldyn?

7) Lines 824, 828: the exponent for rho is missing

**Have all data underlying the figures and results presented in the manuscript been provided?**

Reviewer #1: **No: **The manuscript directs readers to a github site with the code used the generate the data and figures. However, as of January 4, that repository is currently empty.

Reviewer #2: Yes

PLOS authors have the option to publish the peer review history of their article (what does this mean?). If published, this will include your full peer review and any attached files.

Reviewer #1: **Yes: **Steven S. Andrews

Reviewer #2: No
---

## [Decision Letter · Decision Letter 1]

24 Jun 2021

Dear Dr. Ramirez,

We are pleased to inform you that your manuscript 'A novel stochastic simulation approach enables exploration of mechanisms for regulating polarity site movement' has been provisionally accepted for publication in PLOS Computational Biology.

Best regards,

Attila Csikász-Nagy

Associate Editor

PLOS Computational Biology

Mark Alber

Deputy Editor

PLOS Computational Biology

Reviewer's Responses to Questions

**Comments to the Authors:**

Reviewer #1: The authors have fully addressed all of my concerns. I recommend publication as is.

Reviewer #2: The authors have performed extensive revisions that improved the manuscript and addressed my comments.

I have just one comment regarding the definition of reaction versus diffusion limits (for examples lines 764-776) that the authors may consider when submitting the final version of the paper. I believe the added definition of reaction- versus diffusion-limited in the paper is consistent with common use of these terms in 3D systems. In 2D, however, even in cases where D >> lambda rho^2 (classified as reaction-limit in the paper), the system may still become diffusion-limited over long times: unlike in 3D where diffusion is a dilute exploration of space, in 2D diffusion brings the same particles together with increased probability over time (albeit only logarithmically so). So after some cross-over time/distance, any small lambda system may be driven to become diffusion-controlled in the true sense (i.e. limited by diffusion over distances larger than molecular size, not just diffusion over the molecular size). Thus, in principle, it’s not just the case of D << lambda rho^2 (classified as diffusion-limited in the paper) that may be complicated but possibly also D >> lambda rho^2. However this may not be a problem if the simulation methods already account for the D << lambda rho^2 limit.

**Have the authors made all data and (if applicable) computational code underlying the findings in their manuscript fully available?**

Reviewer #1: Yes

Reviewer #2: Yes

PLOS authors have the option to publish the peer review history of their article (what does this mean?). If published, this will include your full peer review and any attached files.

Reviewer #1: **Yes: **Steven S. Andrews

Reviewer #2: No

---

## [Editor Report · Acceptance letter]

6 Jul 2021

PCOMPBIOL-D-20-02174R1 

A novel stochastic simulation approach enables exploration of mechanisms for regulating polarity site movement

Dear Dr Ramirez,

I am pleased to inform you that your manuscript has been formally accepted for publication in PLOS Computational Biology. Your manuscript is now with our production department and you will be notified of the publication date in due course.

With kind regards,

Olena Szabo
